# Longitudinal symptom dynamics of COVID-19 infection

Barak Mizrahi [1,8], Smadar Shilo [2,3,4,8], Hagai Rossman[2,3,8], Nir Kalkstein [1], Karni Marcus[1,7], Yael Barer[5], Ayya Keshet[2,3], Na'ama Shamir-Stein[6], Varda Shalev[7], Anat Ekka Zohar[6], Gabriel Chodick [5,7] & Eran Segal [2,3 ✉]

As the COVID-19 pandemic progresses, obtaining information on symptoms dynamics is of essence. Here, we extracted data from primary-care electronic health records and nationwide distributed surveys to assess the longitudinal dynamics of symptoms prior to and throughout SARS-CoV-2 infection. Information was available for 206,377 individuals, including 2471 positive cases. The two datasources were discordant, with survey data capturing most of the symptoms more sensitively. The most prevalent symptoms included fever, cough and fatigue. Loss of taste and smell 3 weeks prior to testing, either self-reported or recorded by physicians, were the most discriminative symptoms for COVID-19. Additional discriminative symptoms included self-reported headache and fatigue and a documentation of syncope, rhinorrhea and fever. Children had a significantly shorter disease duration. Several symptoms were reported weeks after recovery. By a unique integration of two datasources, our study shed light on the longitudinal course of symptoms experienced by cases in primary care.

[1] KI Research Institute, Kfar Malal, Israel. [2] Department of Computer Science and Applied Mathematics, Weizmann Institute of Science, Rehovot, Israel. [3] Department of Molecular Cell Biology, Weizmann Institute of Science, Rehovot, Israel. [4] Pediatric Diabetes Unit, Ruth Rappaport Children's Hospital, Rambam Healthcare Campus, Haifa, Israel. [5] Maccabi Institute for Research and Innovation, Tel Aviv, Israel. [6] Quality, Research and Evaluation Administration, Maccabi Healthcare Services, Tel Aviv, Israel. [7] Sackler Faculty of Medicine, Tel-Aviv University, Tel-Aviv, Israel. [8] These authors contributed equally: Barak Mizrahi, Smadar Shilo, Hagai Rossman. ✉email: eran.segal@weizmann.ac.il

In December 2019 a cluster of severe respiratory disease from an unknown cause was identified in Wuhan, China. Soon after, the causative pathogen was identified as a novel coronavirus and was named the severe acute respiratory syndrome coronavirus 2 (SARS-CoV-2)[1]. Since then, the virus has rapidly spread across China and worldwide, affecting by July 2020 more than 12,000,000 confirmed patients and 500,000 deaths worldwide in above 200 countries, areas or territories with confirmed cases, creating a major global health crisis[2].

As the virus spread rapidly across the globe, causing an increased number of infected individuals, reports on the clinical characteristics of the disease have started to emerge. Fever, cough, myalgia, and fatigue were described as common symptoms of the disease. Less common symptoms included sputum production, headache, haemoptysis, and diarrhea[3–8]. Further on, anosmia and ageusia also emerged as prevalent and relatively discriminative symptoms of COVID-19 infection[9–12]. As clinical data started to accumulate, reviews describing the potential cardiovascular[13], gastrointestinal[14], neurological[15], and cutaneous manifestations[16] of the disease were published.

The majority of studies published to date describing the clinical course of patients with COVID-19 infection were based on retrospective data of adult hospitalized patients[3–8]. This creates a knowledge gap, as infected people who only experience mild symptoms, do not necessarily seek clinical care. The clinical course of COVID-19 in children is less described, with more than 90% of all pediatric patients were previously described as asymptomatic, mild, or moderate[17], Information regarding the dynamic of symptoms throughout the disease course and their overall duration in both children and adults is still lacking.

In Israel, the first infection of COVID-19 was confirmed on February 21st, 2020. In response, the Israeli Ministry of Health (MOH) employed a series of physical distancing measures in an attempt to mitigate the spread of the virus, which were later on gradually relieved. Throughout the pandemic, testing policy in Israel has changed. Initially, only individuals who fulfilled both clinical criteria (symptoms of fever or respiratory symptoms) and epidemiological criteria (such as being in proximity to COVID-19 patients) were tested but further on, more indications for testing were added[18]. Here, we analyzed a unique dataset composed of electronic health records (EHR) from Maccabi Health Services (MHS), the second largest Health Maintenance Organization (HMO) in Israel which includes the results of SARS-CoV-2 PCR testing and primary care visits, and linked longitudinal self-reported symptoms reported as part of a nationwide survey[19], to better understand the full clinical spectrum of symptoms experienced by adults and children infected with COVID-19.

## Results

From 1/3/2020 to 07/06/2020, information on symptoms was available for 206,377 individuals (see "Methods" section). Information on symptoms was obtained from two sources: 117,230 individuals performed a PCR test for SARS-CoV-2 and had a record of primary care visit (Table 1). Of them, 52,298 had a primary care visit with documented symptoms (2214 positive, 50,084 negative). 159,162 individuals (499 positive, 10,984 negative and 147,679 not tested) filled at least one-self-reported symptoms survey (Table 1). 5083 individuals had information on symptoms from both sources. A snapshot of the detailed clinical course for one individual with COVID-19 infection in our cohort is presented as an example in Fig. 1.

Altogether, information on symptoms was available for 2471 individuals defined as positive COVID-19 cases, 56,227 defined as negative and 147,679 individuals who had no record of a PCR test for SARS-CoV-2. Among the positive cases, and throughout the

study period, 508 (20.6%) individuals were hospitalized, including 51 (2.1%) individuals that were admitted to the Intensive care unit (ICU) and 16 (0.6%) patients died.

First, we analyzed the similarity between self-reported symptoms and EHR-captured data. From the 5,083 participants who had information on symptoms from both data sources, we identified 915 different events for a total of 706 individuals in which the same person filled a self-reported survey and had a physician visit documented in the EHR on the same day. When comparing these events, we found the overall agreement between the two sources was generally low. Overall, most of the symptoms, with the exception of fever and myalgia, were self-reported in a higher percentage than they were recorded by physicians in the EHR (Supplementary Table 1). As expected, symptoms which are part of the Israeli testing policy had a higher agreement between the two sources since they were more likely to be asked by a physician during the visit. These included cough, which had a 52% agreement between the two sources and fever, which had a 34% agreement. Diarrhea also had a relatively high agreement of 35%. Other symptoms had a lower agreement of up to 16%. Disturbance of the sensation in smell and taste, had no agreement at all between the two sources, and were mostly self-reported, potentially due to the fact that early in the course of the pandemic the evidence on the existence of this symptoms in individuals infected with COVID-19 was not strong, so it is possible that it was less asked and reported by physicians. Due to the fact that the majority of individuals did not have information from the two sources of information, the different characteristics of each of the population (Table 1), the potential biases in each of the databases and the low agreement between the two sources of information in this subgroup, we decided to analyze symptoms separately for each source of information.

**Clinical manifestations of COVID-19 infection in adults**. Overall, the most prevalent symptoms recorded in primary care visits throughout the study period in adult COVID-19 cases were cough (11.6%), Fever (10.3%), Myalgia (7.7%), and Fatigue (5.9%). Emotional disturbance, including anxiety and depression, were common (15.9%). The most prevalent self-reported symptoms were cough (21%), fatigue (19%), rhinorrhea, and/or nasal congestion (17%), headache (16%), and myalgia (11%) (Table 1). Of note, in order to obtain the full clinical picture, responders to the survey were asked whether they experience additional symptoms and were given an option to report these in length and in a free text format. Only few of the positive COVID-19 cases reported additional symptoms and those were mainly reported before disease onset. These included: abdominal pain, chest discomfort/chest pain, loss of appetite, bitter taste and chills.

**Clinical manifestations of COVID-19 infection in children**. A total of 21,567 children were included in the analysis. Of them, 862 (4%) were positive COVID-19 cases (mean age of 10.69 ± 5.09 years old) and 20,705 (96%) negative (mean age of 8.67 ± 5.46 years old). The percentage of positive cases from all individuals tested was similar between children and adults (4% versus 4.25%, respectively). Data on clinical symptoms of these children was obtained solely from the MHS EHR as the survey was not distributed at this age group. Conjunctivitis, rash, sore throat, dyspnea and/or shortness of breath and speech disturbance, had a higher prevalence in children who were positive to COVID-19 compared to positive adults. Overall, the most prevalent symptoms recorded in primary care visits throughout the study period included fever (7%), cough (5.5%), abdominal pain (2.4%), and fatigue (2.3%). Emotional disturbance, including anxiety and depression, were less documented in children compared to adults,

**Table 1 Baseline characteristics of the study cohort.**

**A. Individuals with primary care visits**

| Characteristic, mean (SD) or % | All individuals n = 117,230 (100%) | Adults (n = 95,663) | | Children (n = 21,567) | |
|---|---|---|---|---|---|
| | | COVID-19 negative n = 91,597 (78.1%) | COVID-19 positive n = 4066 (3.5%) | COVID-19 negative n = 20,705 (17.7%) | COVID-19 positive n = 862 (0.7%) |
| Age in years | 39.30 (22.83) | 46.32 (19.38) | 43.03 (18.34) | 8.67 (5.46) | 10.69 (5.09) |
| Gender (male) | 51,820 (44%) | 38,185 (42%) | 2273 (56%) | 10,924 (53%) | 438 (51%) |
| &Presence of a chronic medical condition | 55,903 (48%) | 50,994 (56%) | 1962 (48%) | 2829 (14%) | 118 (14%) |
| *Number of primary care visits with documented symptoms* | | | | | |
| Total | 120,120 | 98,312 | 4894 | 16,530 | 384 |
| *Prior to COVID-19 test | 65,097 | 51,793 | 1436 | 11,702 | 166 |
| *After COVID-19 test | 55,023 | 46,519 | 3458 | 4828 | 218 |
| *Symptoms recorded in primary care visits* | | | | | |
| Abdominal pain | 5988 (5.1%) | 5103 (5.6%) | 154 (3.8%) | 710 (3.4%) | 21 (2.4%) |
| Arthralgia | 4338 (3.7%) | 4034 (4.4%) | 132 (3.2%) | 163 (0.8%) | 9 (1.0%) |
| Chest pain or discomfort | 3372 (2.9%) | 3069 (3.4%) | 185 (4.5%) | 111 (0.5%) | 7 (0.8%) |
| Conjunctivitis | 1171 (1.0%) | 752 (0.8%) | 19 (0.5%) | 383 (1.8%) | 17 (2.0%) |
| Constipation | 975 (0.8%) | 795 (0.9%) | 19 (0.5%) | 157 (0.8%) | 4 (0.5%) |
| Cough | 13,427 (11.5%) | 10,179 (11.1%) | 470 (11.6%) | 2731 (13.2%) | 47 (5.5%) |
| Diarrhea | 2255 (1.9%) | 1655 (1.8%) | 68 (1.7%) | 519 (2.5%) | 13 (1.5%) |
| Disturbance of skin sensation | 678 (0.6%) | 657 (0.7%) | 16 (0.4%) | 5 (0.0%) | 0 (0.0%) |
| Disturbances of sensation of smell and taste | 264 (0.2%) | 203 (0.2%) | 44 (1.1%) | 15 (0.1%) | 2 (0.2%) |
| Dizziness | 1665 (1.4%) | 1542 (1.7%) | 61 (1.5%) | 60 (0.3%) | 2 (0.2%) |
| Dyspnea and or shortness of breath | 1385 (1.2%) | 778 (0.8%) | 41 (1.0%) | 553 (2.7%) | 13 (1.5%) |
| Emotional disturbance | 8209 (7.0%) | 7154 (7.8%) | 648 (15.9%) | 371 (1.8%) | 36 (4.2%) |
| Fatigue | 4284 (3.7%) | 3713 (4.1%) | 240 (5.9%) | 311 (1.5%) | 20 (2.3%) |
| Fever | 11,093 (9.5%) | 6467 (7.1%) | 420 (10.3%) | 4146 (20.0%) | 60 (7.0%) |
| General symptoms (Amnesia, chills, generalized pain or hypothermia) | 628 (0.5%) | 589 (0.6%) | 19 (0.5%) | 20 (0.1%) | 0 (0.0%) |
| Headache | 3318 (2.8%) | 2794 (3.1%) | 126 (3.1%) | 387 (1.9%) | 11 (1.3%) |
| Hearing loss | 1063 (0.9%) | 811 (0.9%) | 31 (0.8%) | 219 (1.1%) | 2 (0.2%) |
| Lymphadenopathy | 448 (0.4%) | 362 (0.4%) | 11 (0.3%) | 73 (0.4%) | 2 (0.2%) |
| Myalgia | 9900 (8.4%) | 9202 (10.0%) | 314 (7.7%) | 375 (1.8%) | 9 (1.0%) |
| Nausea and or vomiting | 1167 (1.0%) | 870 (0.9%) | 49 (1.2%) | 241 (1.2%) | 7 (0.8%) |
| Neuralgia | 426 (0.4%) | 411 (0.4%) | 12 (0.3%) | 3 (0.0%) | 0 (0.0%) |
| Palpitation | 817 (0.7%) | 761 (0.8%) | 42 (1.0%) | 13 (0.1%) | 1 (0.1%) |
| Rash | 1127 (1.0%) | 585 (0.6%) | 27 (0.7%) | 495 (2.4%) | 20 (2.3%) |
| Runny nose and or nasal congestion | 1293 (1.1%) | 942 (1.0%) | 49 (1.2%) | 297 (1.4%) | 5 (0.6%) |
| Sleep disturbance | 1049 (0.9%) | 925 (1.0%) | 47 (1.2%) | 75 (0.4%) | 2 (0.2%) |
| Sore throat | 1378 (1.2%) | 1100 (1.2%) | 30 (0.7%) | 241 (1.2%) | 7 (0.8%) |
| Speech disturbance | 355 (0.3%) | 145 (0.2%) | 7 (0.2%) | 198 (1.0%) | 5 (0.6%) |
| Syncope | 529 (0.5%) | 450 (0.5%) | 24 (0.6%) | 52 (0.3%) | 3 (0.3%) |
| Tachycardia | 221 (0.2%) | 183 (0.2%) | 23 (0.6%) | 13 (0.1%) | 2 (0.2%) |
| Voice disturbance | 285 (0.2%) | 254 (0.3%) | 14 (0.3%) | 16 (0.1%) | 1 (0.1%) |
| Weight loss | 600 (0.5%) | 546 (0.6%) | 18 (0.4%) | 35 (0.2%) | 1 (0.1%) |

**B. survey responders**

| Characteristic, mean (SD) or % | All individuals n = 159,162 (100%) | Individuals not tested for COVID-19 N = 147,679 (92.78%) | COVID-19 negative n = 10,984 (6.9%) | COVID-19 positive n = 499 (0.31%) |
|---|---|---|---|---|
| Age in years | 46.62 (16.94) | 46.84 (16.95) | 43.82 (16.45) | 42.15 (16.61) |
| Gender (male) | 66,777 (42%) | 62,461 (42%) | 4085 (37%) | 231 (46%) |
| &Presence of a chronic medical condition | | | 5789 (53%) | 231 (46%) |
| Healthcare workers | 9934 (6%) | 8279 (6%) | 1602 (15%) | 53 (11%) |
| *Self-reported surveys* | | | | |
| Total | 1,262,479 | 1,187,523 | 72,928 | 2028 |
| *Prior to COVID-19 test | | | 36,169 | 371 |
| *After COVID-19 test | | | 36,759 | 1657 |
| Per individual | 7.93 (9.96) | 8.04 (10.05) | 6.64 (8.72) | 4.06 (5.98) |
| Mean (SD) | 3 (1–11) | 3 (1–11) | 2 (1–8) | 2 (1–4) |
| Median(IQR) | | | | |

**Table 1 (continued)**

**B. survey responders**

| Characteristic, mean (SD) or % | All individuals n = 159,162 (100%) | Individuals not tested for COVID-19 N = 147,679 (92.78%) | COVID-19 negative n = 10,984 (6.9%) | COVID-19 positive n = 499 (0.31%) |
|---|---|---|---|---|
| *Self-reported symptoms* | | | | |
| Chills | 2430 (2%) | 1951 (1%) | 467 (4%) | 12 (2%) |
| Confusion | 1364 (1%) | 1143 (1%) | 206 (2%) | 15 (3%) |
| Cough | 20,162 (13%) | 17,374 (12%) | 2682 (24%) | 106 (21%) |
| Diarrhea | 6889 (4%) | 6060 (4%) | 804 (7%) | 25 (5%) |
| Dry cough | 11,340 (7%) | 9586 (6%) | 1676 (15%) | 78 (16%) |
| Fatigue | 12,630 (8%) | 10,745 (7%) | 1792 (16%) | 93 (19%) |
| Feel well | 128,633 (81%) | 120,424 (82%) | 7860 (72%) | 349 (70%) |
| Fever (body temperature above 38 °C) | 863 (1%) | 602 (0%) | 250 (2%) | 11 (2%) |
| Headache | 19,818 (12%) | 17,585 (12%) | 2152 (20%) | 81 (16%) |
| Loss of taste or smell | 1209 (1%) | 951 (1%) | 209 (2%) | 49 (10%) |
| Myalgia | 9011 (6%) | 7766 (5%) | 1192 (11%) | 53 (11%) |
| Nausea and/or vomiting | 3793 (2%) | 3290 (2%) | 488 (4%) | 15 (3%) |
| Other symptoms | 11,066 (7%) | 10,015 (7%) | 1004 (9%) | 47 (9%) |
| Rhinorrhea and/or nasal congestion | 25,828 (16%) | 23,205 (16%) | 2536 (23%) | 87 (17%) |
| Shortness of breath | 4248 (3%) | 3515 (2%) | 709 (6%) | 24 (5%) |
| Sore throat | 13,684 (9%) | 11,766 (8%) | 1870 (17%) | 48 (10%) |
| Wet cough | 11,509 (7%) | 9978 (7%) | 1482 (13%) | 49 (10%) |

Prevalence of symptoms that were present in more than 10 positive cases throughout the study period is presented in A individuals with primary care visits and B self-reported symptoms.
*Number of primary care visits/ filled surveys throughout the study period, prior to and after the PCR test. Date of the test was considered as the first positive test for the positive COVID-19 cases and the first negative test for the negative cases.
#Chronic medical conditions that were included in the analysis are cardiovascular diseases, diabetes, hypertension, obesity, underweight, malignancy, cystic fibrosis, chronic renal failure and dialysis treatments, chronic obstructive pulmonary disease, depression, osteoporosis, inflammatory bowel disease, coagulation, blood disorder and warfarin treatments, cognitive impairment and the need for special home therapies.

but were present in 4.2% of positive cases. Most symptoms were rare, and were present in less than 1% of the patients.

**Dynamics of symptoms in COVID-19 patients**. We next analyzed the dynamics of symptoms reported by individuals with confirmed COVID-19 infection compared to individuals with negative COVID-19 tests in time. To that end, we made use of the rich individual level data obtained from both of the data sources we had in our possession: the self-reported symptoms and the symptoms which were recorded in the EHR during primary care visits. These data were analyzed separately (see "Methods" section).

A longitudinal analysis of symptoms on the entire cohort from both data sources and in regard to the time of test and time of recovery revealed different patterns in time for different symptoms (Fig. 1). Interestingly, there is an increase in loss of taste and smell sensation in COVID-19 patients, as compared to negative cases, which is apparent both in the self-reported surveys, and in a smaller magnitude also in primary care visits. The prevalence of these symptoms start increasing as early as 3 weeks prior to time of diagnosis, decreasing up to approximately a week prior to recovery date.

As fever was one of the main criteria to fulfill in order to be applicable for COVID19 test in Israel in the majority of time during the COVID-19 pandemic, it is not surprising that we observe an increase in this symptom in both positive and negative cases with a peak just before the time of testing, and that this symptom is more evident in primary care visits. Nonetheless, the percentage of fever in positive cases is still higher than the negative cases, both in the survey and in primary care visits. Similarly, cough was also a part of the criteria for COVID19 testing and also had a peak in prevalence just before the time of testing for both positive and negative cases.

Fatigue was also self-reported by participants in a higher prevalence around the time of testing in both positive and

negative cases, but with a higher prevalence in the positive cases. The dynamics of sore throat and diarrhea was similar in positive and negative cases with the maximal prevalence reported around the diagnostic test. Higher prevalence of these symptoms was documented by physicians in negative cases at time of diagnosis. self-reported headache was more prevalent in positive cases, manifest in up to 30% of patients prior to the day of diagnosis, and gradually decreasing up to 2 weeks before recovery when it is diminished. The dynamics of additional symptoms which were reported by individuals in our cohort or recorded by their physicians is presented in Fig. 2.

As symptoms may be differentially present throughout the disease course, a time-dependent analysis is warranted. Accordingly, we construct Kaplan–Meier curves from the time in which the symptom is self-reported or recorded in the EHR to a positive PCR result. Hazard ratios (HR) for each symptom, calculated by Cox proportional hazards models, and adjusted for age, gender, presence of a chronic medical condition and time (number of days since study initiation) are presented in Fig. 2 to account for the time from symptoms onset to COVID-19 testing results. We present these results separately for self-reported symptoms and for EHR recorded symptoms. The importance of loss of taste or smell symptoms is demonstrated in these analyses, with an HR of 22.5 and 13.3 followed by fever with HR of 9.7 and 4.3 in self-reported symptoms and EHR, respectively. In addition, several symptoms such as nausea or vomiting, muscle pain, and headache reveal different patterns between the two sources of information. Individuals who self-reported these symptoms had an increasingly higher percentage of positive tests in contrast to their record in the EHR (Fig. 2, full results are presented in Supplementary note 6, Supplementary Tables 6 and 7).

**Clinical manifestations of COVID-19 infection after recovery**. Data on symptoms after recovery were obtained from both data

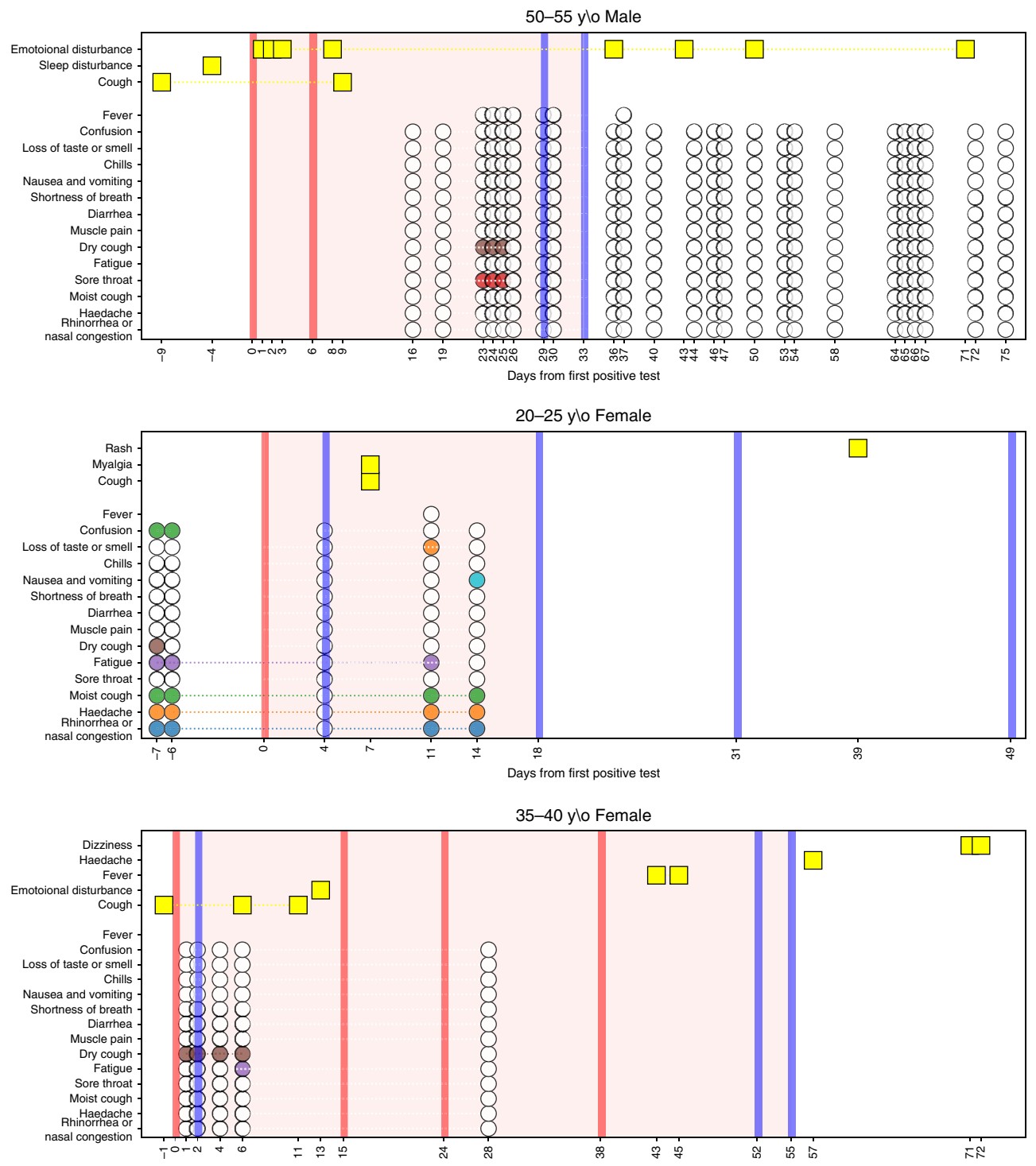

**Fig. 1 Longitudinal dynamics of symptoms in infected individuals.** Three examples are given. Yellow rectangles represent symptoms recorded by a physician at a primary care visit. Colored circles represent symptoms self-reported by the individual throughout the survey. White circles represent self-report of not experiencing a symptom through the survey. Red and blue vertical lines indicate a positive or negative PCR test for SARS-CoV-2, respectively. The area marked in light red indicates the period of time an individual was considered as infected with COVID-19.

sources. From 2214 positive cases who had a record of primary care visit with symptoms, 1818 recovered throughout the study period, and from them, 909 had a primary care visit after recovery, with average follow up time of 31.4 ± 20.1 days after recovery.

From 499 positive cases who filled at least one-self-reported symptoms survey, 442 recovered, 278 of them have filled a survey after recovery with an average follow up time of 17.6 ± 14.8 days. Long duration of symptoms, specifically fatigue, myalgia, runny nose and shortness of breath was observed weeks after recovery.

**Variability in the clinical course of COVID-19 infection.** After analyzing the general dynamics of symptoms, we further looked

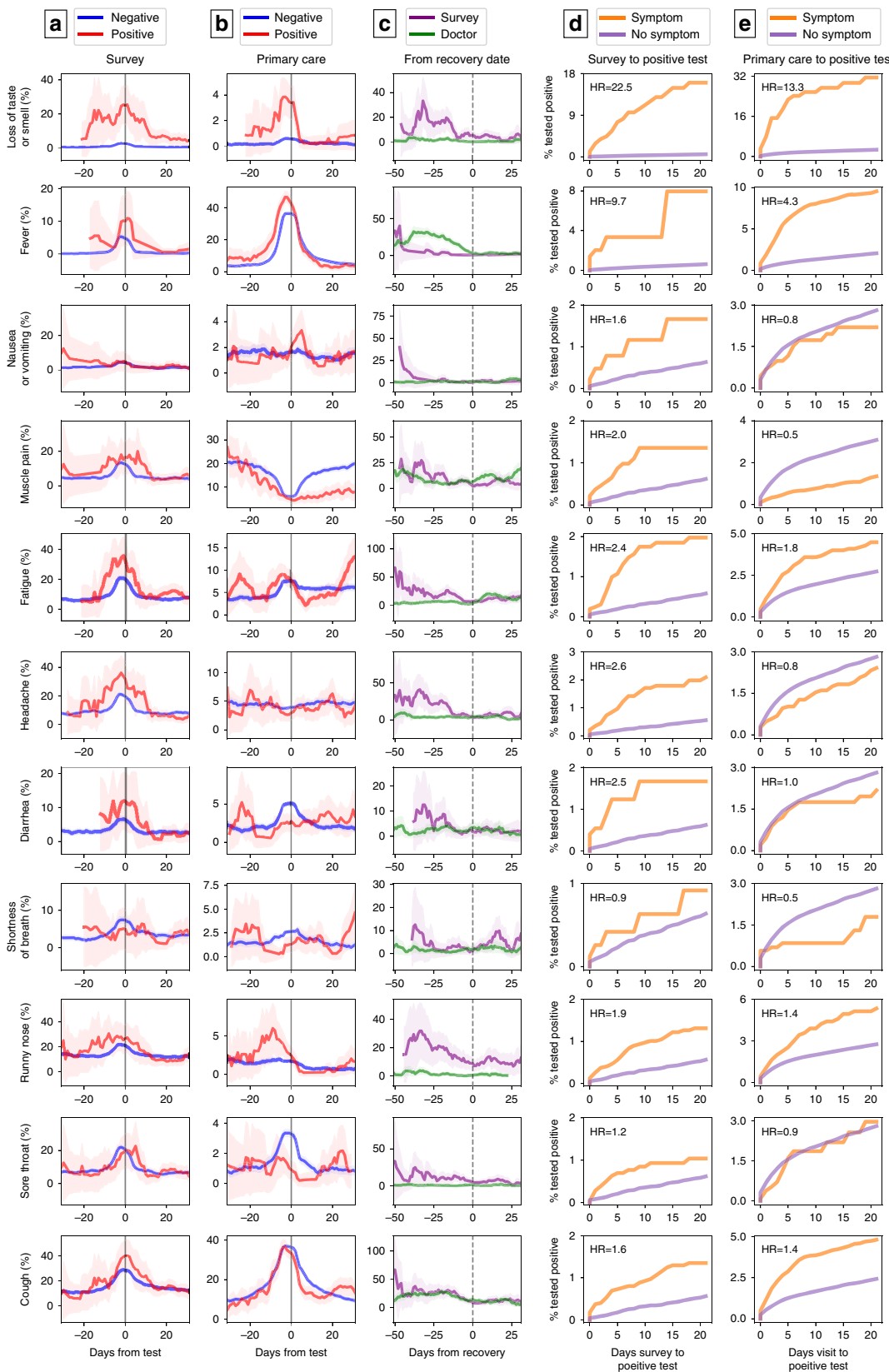

at the variability of the clinical course in different individuals in the cohort. A high variability in the clinical course was observed by two physicians who reviewed the medical charts and surveys of COVID-19 cases. Clinical spectrum was ranging from mild and short disease to a prolonged course, lasting weeks after the recovery. Several examples are shown in Fig. 1. Among recovered individuals, mean disease duration was $23.5 \pm 9.9$ days ($n = 2045$), thus showing a high variability of disease duration. Cox regression analyses revealed that children have a significantly shorter recovery time compared to adults (HR = 1.18 (1.01–1.39),

**Fig. 2 Dynamics of symptoms in COVID-19 patients.** Columns (**a–c**): each row represents the prevalence of a symptom in our cohort analyzed by time relative to diagnosis day from (**a**): Survey of self-reported symptoms and (**b**): Primary care visits in positive COVID19 cases (red) versus negative (blue) (**c**): Prevalence of symptoms in our cohort relative to time of recovery by surveys of self-reported symptoms (purple) and primary care visits (green). Each time point is calculated by taking a 1 week window (±3 days from day). Error bands represent 95% binomial proportion confidence intervals. Columns **d**, **e**: Kaplan–Meier curves from the time in which a symptom is self-reported (**d**) or recorded in the EHR (**e**) to a positive PCR result. The curves present the cumulative incidence of individuals who tested positive for COVID-19 and have a specific symptom (orange) versus those who did not report this symptom (purple) in time. Hazard ratios (HR) adjusted for gender, age, prior conditions and time (number of days since study initiation) are indicated. Note the *y*-axis scale is different for each panel.

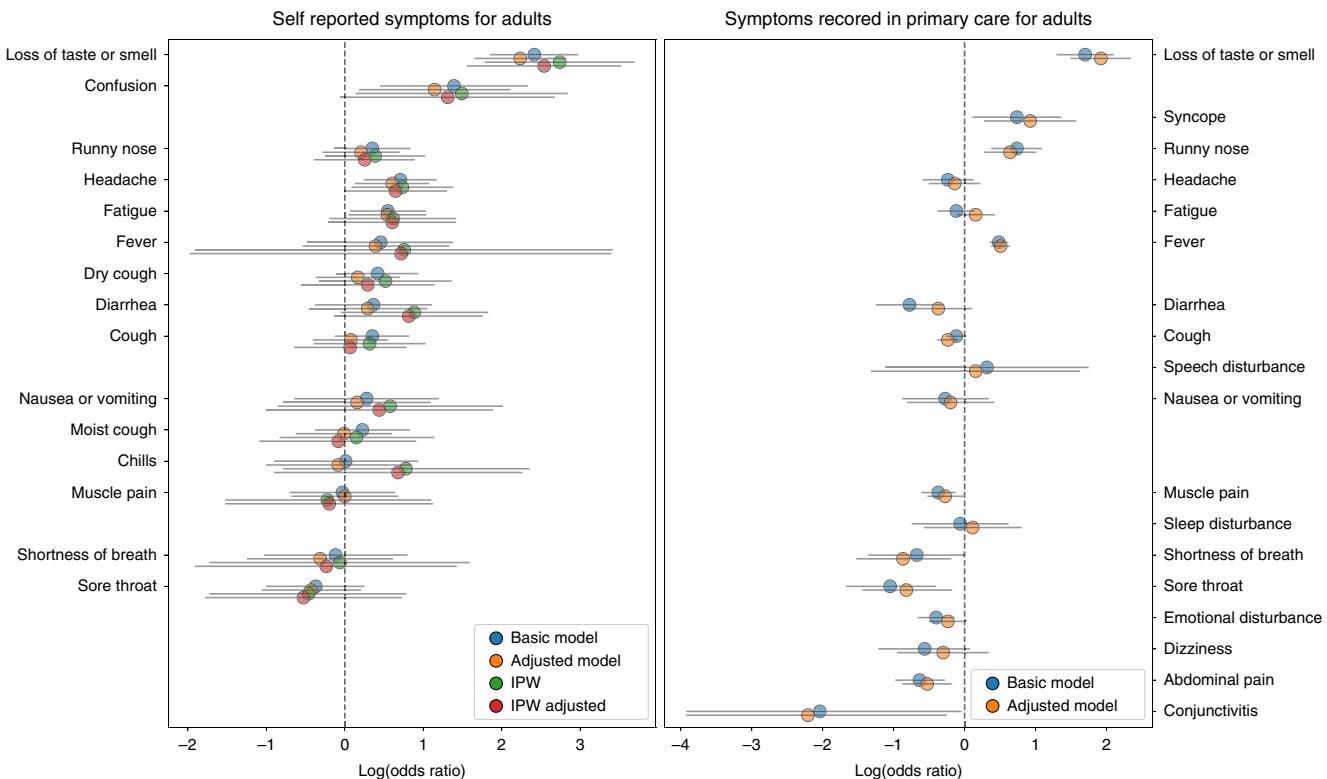

**Fig. 3 Odds ratio analysis of symptoms prior to COVID-19 test.** Log odds ratio (OR) calculated by the prevalence of symptoms 21 days prior to COVID-19 test for adults in self recorded symptoms (*n* = 4843) versus EHR-captured symptoms (*n* = 70,606). Calculated log odds ratios are presented along with gray lines indicating 95% confidence intervals. Blue circles represent log OR calculated from the basic model, orange circles represent log OR from the adjusted model, green circles represent log OR after applying Inverse probability weighting (IPW) to help address biases related to those receiving testing and red circles represents log OR from the adjusted model after applying IPW.

*p* = 0.04). Gender and the presence of a chronic medical condition did not affect the recovery time significantly (*p* = 0.46 and *p* = 0.87, respectively). Next, we analyzed whether the presence of specific symptoms was associated with different time to recovery. This analysis revealed that for example, individuals with loss of smell and taste tend to have a shorter time to recovery compared to those experiencing shortness of breath, possibly due to the fact that the latter represents a disease in higher severity level (see Supplementary note 4, Supplementary Fig. 2).

**Associations between symptom reports and positive tests.** In order to distill the symptoms that can assist physicians in identifying COVID-19 patients prior to diagnosis, we next analyzed the association between symptoms experienced by individuals in the 3 weeks prior to the time of testing and COVID-19 diagnosis and were present in at least 10 positive cases (Fig. 3). This analysis was done separately for children and adults. For adults, loss of taste or smell, either self-reported or documented by physician, were the symptoms which were most associated with a positive diagnosis (odds ratio (OR) = 11.18; 95% confidence interval (CI)

6.43–19.44 and OR = 5.47 (3.69–8.09) for self-reported and primary care visit documentation, respectively).

self-reported headache (OR = 2.03 (1.29–3.19)) and fatigue (OR = 1.73 (1.08–2.79) were also significantly associated with COVID-19 diagnosis, as well as syncope (OR = 2.09 (1.13–3.88)), runny nose (OR = 2.09 (1.47–2.95)) and fever (OR = 1.62 (1.44–1.83)) documented by a physician in a primary care visit. self-reported confusion (OR = 4.02 (1.58–10.21) was also associated with positive cases but the first was reported by only 5 individuals in the 3 weeks prior to the diagnostic test. In contrast, self-report of a body temperature below 37.4 °C and a physician documentation of myalgia, diarrhea, sore throat, abdominal pain, shortness of breath and conjunctivitis were significantly associated with individuals negative to COVID-19 (OR of 0.53 (0.34–0.80), 0.69 (0.55–0.87), 0.46 (0.29–0.74), 0.35 (0.19–0.66), 0.53 (0.38–0.75), 0.51(0.26–0.98), and 0.13(0.02–0.95), respectively). Results on self reports for these symptoms were only partially available and in some cases revealed an opposite trend, although not statistically significant.

We next adjusted OR by age, gender, presence of a chronic medical condition and time and applied Inverse probability

weighting (IPW) for self-reported symptoms to help address biases related to those receiving testing (see "Methods" section). After OR adjustment, the same symptoms remained significantly associated with a positive COVID-19 diagnosis. After applying IPW and the adjusted IPW, these symptoms also remain significant with the exception of fatigue. In children, sample size was smaller and information on symptoms was only available from the EHR. Only 3 symptoms were experienced in the 3 weeks prior to the diagnostic test by more than 10 positive individuals: cough, fever and fatigue. Fever (OR = 0.3 (0.22–0.42)) and cough (OR = 0.4 (0.28–0.59)) were associated with negative COVID-19 cases. Analysis of the less prevalent symptoms revealed that syncope and loss of taste and smell had the highest OR for COVID-19 infection (2.45 for both) but were not statistically significant.

Full results of OR analyses are presented in Supplementary note 5.

## Discussion

In this study, we leveraged a unique dataset that includes EHR of primary care visits from the second largest HMO in Israel, and linked longitudinal self-reported symptoms surveys collected prior to and after COVID-19 testing. These two sources of information enable us to comprehensively capture data on symptoms of mostly mild COVID-19 cases from two different perspectives: the reports of the patient themself, and the reports of their physician. As previously reported, we observed a high variability in the clinical course of COVID-19 patients. Recovery time was highly variable, with a mean duration of 23.5 ± 9.9 days, and was significantly shorter in children (p-value = 0.04).

Our study shed light on the prevalence of clinical symptoms experienced in mild COVID-19 patients. Previous studies which were mostly based on hospitalized patients have marked fever (90% or more), cough (around 75%), and dyspnea (up to 50%), as well as gastrointestinal symptoms in a small subset of patients to be the main clinical manifestations of COVID-19[20]. In our cohort, consisting of primarily non-hospitalized patients with mild disease, the prevalence of these symptoms was substantially lower. The most prevalent symptoms in adults recorded in EHR were cough (11.6%), fever (10.3%), myalgia (7.7%), and fatigue (5.9%) and the most prevalent self-reported symptoms were cough (21%), fatigue (19%), and rhinorrhea and/or nasal congestion (17%). In children, the most prevalent symptoms recorded in the EHR were fever (7%), cough (5.5%), abdominal pain (2.4%), and fatigue (2.3%). Temporal dynamics of self-reported and documented symptoms revealed different patterns of symptoms along time, and long duration of symptoms, specifically fatigue, myalgia, runny nose and shortness of breath, which were reported by individuals weeks after recovery. These findings, inline with a previous report by Tenforde et al.[21], demonstrates that COVID-19 infection may result in prolonged symptoms, even among outpatient individuals with mild disease.

Disturbances of the sensation of smell and taste was documented by physicians in 1.1% of the adult patient and 0.2% of the children but was self-reported in 10% of the cases. Although these symptoms were not common, they emerged in our study as having the highest HR and OR for COVID-19 diagnosis when appearing 3 weeks prior to COVID-19 tests in both adults and children (HR of 22.5 and 13.3 as a self-reported symptom and EHR, respectively, OR = 11.18 and OR = 5.47 as self-reported or documented by physician for adults, and OR = 2.45 for children). In children, the result was not statistically significant, possibly due to a small sample size. Although anosmia and ageusia were initially less described as COVID-19 symptoms[3,6], and were not part of the Israeli testing policy throughout the study period, they further emerged as the most

predictive symptoms for COVID-19 in this study and others[10,12,22], and were also found to be the most prolong symptoms in infected individuals[23]. Health organizations worldwide have gradually added it to the list of COVID-19 related symptoms including the USA CDC[24], the UK[25] and Israel[26].

Other significant self-reported symptoms which were reported by at least 10 positive COVID-19 cases in the 3 weeks prior to testing and are still not included in formal testing policies were headache (OR = 2.03) and fatigue (OR = 1.73), which was not statistically significant after applying IPW and adjusted IPW. These symptoms are not specific to COVID-19, but may potentially help discriminate between positive and negative cases. Interestingly, syncope documented by a physician 3 weeks prior to diagnosis was significantly higher in positive cases in adults (OR = 2.09) and had a high OR but not statistically significant in children (OR = 2.45). Nonetheless, the overall prevalence of this symptom was very low in the cohort (0.6% and 0.3% of positive adults and children, respectively) and although it was described as an atypical presentation of COVID-19 infection[27,28], reports on syncope as a COVID-19 related symptom thus far were sparse[28]. Therefore, more studies are needed in order to determine if this symptom is part of COVID-19 clinical sequelae. In children, several symptoms, including some which are part of the Isreali testing guidelines, such as cough (OR = 0.4) and fever (OR = 0.3) were associated with negative cases, possibly due to the fact that many other infectious diseases which are characterized by these symptoms were more prevalent in this age group during this time period.

Emotional disturbance, including anxiety and depression, were documented in 15.9% of the positive adults and 4.2% of the children, and appeared both prior to and after diagnosis. This emphasizes the fact that as the pandemic increases the risk for psychiatric illness, and in addition to medical care, health care providers have to closely monitor the psychosocial needs of their patients[29]. An example is the establishment of new guidelines for emergency psychological crisis intervention initiated for people affected with COVID-19 by the National Health Commission of China[30].

Our study has several strengths. First, the integration of information on symptoms from two different data sources allowed us to comprehensively explore the clinical course of mild COVID-19 cases. As most of the studies on the clinical characteristics of COVID-19 infection thus far rely mostly on hospitalized patients, this information reveals the heterogeneous clinical spectrum of symptoms experienced by COVID-19 infected individuals in the community setting, and may help physicians to identify individuals at risk. Second, data on the presence of clinical symptoms was available for 982 positive cases from either EHR or surveys up to 21 days prior to COVID-19 diagnosis, thus allowing us to capture symptoms prior to performing the diagnostic test in a relatively high percentage of the cohort (40%). This allowed us to prospectively analyze the dynamics of symptoms throughout the disease course, and not only after the diagnosis was already made. Third, COVID-19 cases were identified directly by a documented record of a positive PCR test in the EHR and were not based on self-reported diagnosis. Moreover, in order to check the validity of our survey, we directly compared self-reported answers on COVID-19 diagnoses in our survey to the documented test in the medical chart and found a very high agreement between the two sources. For example, from those who reported not diagnosed with COVID-19 in the survey, only 0.03% had a record of a positive COVID-19 test in the EHR thus further validating our survey. A relatively low agreement between self-reported symptoms and EHR-captured symptoms was found in a subgroup of individuals who had data from the two sources at the same date, which might

be a result of several reasons including the local practices of diagnosis documentation by physicians, who usually document only the main diagnoses as ICD-9 codes. Moreover, previous studies comparing individual's self report versus clinicians reporting in regard to other diagnosis, have found that self-report is more sensitive to identifying symptoms-based conditions[31]. Altogether, we believe that this point highlights the strength of our study in integrating two different data sources together, each with its own pros and cons, to obtain a more complete picture on symptoms as reflected by both perspectives.

This study also has several limitations. First, the testing policy for COVID-19 in Israel has changed throughout the study period[18]. In the majority of the time, individuals had to present with fever or respiratory symptoms, as well as an appropriate epidemiological context, in order to be tested. This may have a strong effect on the prevalence of these symptoms in individuals who were tested compared to those who were not. In addition, both of our data sources may introduce biases to the data and therefore were analyzed separately by us. Data which is based on the voluntary self-reported symptoms of participants is bound to suffer from selection bias. However, the fact that the surveys were distributed by both emails and text messages from the second largest HMO in Israel, a non-profit organization, to all of its adult members from all socioeconomic groups may reduce this bias. Data originating from EHR may also suffer from biases related to processes within the healthcare system[32] and a bias toward patients with more severe conditions.

In conclusion, in this study we analyzed the clinical course of COVID-19 cases in primary care as captured by two data sources, EHRs and surveys, and showed that there is a limitation associated with using EHR data compared to survey data in capturing the full spectrum of symptoms identified when presenting these options to the individual patient in a survey. This highlights the power of survey derived data to enhance understanding of the evolving COVID-19 pandemic. The study provides additional information on the natural history of mostly mild cases of COVID-19 and may alert physicians for the possibility of infection and direct the need for testing and self-isolation.

## Methods

**Data**. In this study, we utilized data originating from MHS. MHS is the second largest HMO currently active in Israel, representing a quarter of the Israeli population. It contains longitudinal data on over 2.3 million people since 1993, with annual attrition rate lower than 1%. Of note, participation in a medical insurance plan is compulsory in Israel, and by the National Health Insurance Law of 1995, all citizens must join one of four official HMO which are run as not-for-profit organizations and are prohibited by law from denying membership to any Israeli resident. The dataset included extensive demographic data, anthropometric measurements, clinic and hospital diagnoses, medication dispensed, and comprehensive laboratory data from a single central lab[33].

For adults, information on symptoms was obtained from two different data sources: symptoms documented in the EHR as ICD-9 codes and symptoms from a self-reported survey. A link to the electronic survey was distributed by emails and text messages by MHS from 01/04/2020 and throughout the COVID-19 pandemic to all adult members (age above 18 years of age) of MHS. The survey included questions relating to age, gender, prior medical conditions, smoking habits, self-reported symptoms and geographical location (see Supplementary note 2). Each participant was asked to fill the survey once a day. The surveys were linked to the EHR by a unique identifier. For children, symptoms were extracted solely from the EHR, as the surveys were not distributed in this age group.

**Study outcome**. Patients with COVID-19 infection were identified as those having at least one record of a positive SARS-CoV-2 polymerase chain reaction (PCR) test in the MHS EHR. PCR tests for SARS-CoV-2 were obtained from nasopharyngeal swabs. Individuals negative to COVID-19 infection were considered as such if all their laboratory tests for SARS-CoV-2 were negative.

**Study design and population**. To obtain information on symptoms in COVID-19 patients, we used two different data sources: symptoms recorded by physicians as ICD-9 codes and self-reported symptoms from the survey (see Supplementary notes 2 and 3, Supplementary Fig. 1).

First, we analyzed data of individuals in MHS, who had at least one PCR test for SARS-CoV-2 between 01/03/2020 and 07/06/2020, and had at least one primary care visit. During the COVID-19 pandemic, visits were held as in-person visits or telemedicine visits. Overall, 175,994 PCR tests for SARS-CoV-2, for 119,583 individuals, were performed during this time period. To capture diagnostic codes given after the date of the diagnostic PCR test, data from primary care visits were extracted from 01/03/2020 to 20/06/2020. During this time period, 117,230 individuals had a documented primary care visit. The characteristics of these individuals are described in Table 1. In order to distill symptoms that are prevalent in COVID19 cases, we first extracted all ICD-9 codes that were documented by a physician during the study period. From these, ICD-9 codes which represent symptoms that were previously described in COVID19 patients were analyzed (see Supplementary note 3).

Second, we analyzed data of individuals in MHS, who had at least one valid self-reported symptoms survey. Between 01/04/2020 to 7/06/2020, 1,313,595 surveys were filled for 181,798 individuals. From them 51,116 (3.9%) surveys were not fully completed by participants and were excluded from the analysis. A total of 1,262,479 surveys for 159,162 adults were included in the analyses. The characteristics of these individuals are described in Table 1. Self-reported fever was defined as a reported body temperature above 38 °C. Chronic medical disease were defined as one of the following: cardiovascular diseases, diabetes, hypertension, obesity, underweight, malignancy, cystic fibrosis, chronic renal failure and dialysis treatments, chronic obstructive pulmonary disease, depression, osteoporosis, inflammatory bowel disease, coagulation, blood disorder and warfarin treatments, cognitive impairment and the need for special home therapies[34].

5083 individuals in MHS who performed a PCR test for SARS-CoV-2, had both a documented primary care visit with recorded symptoms and filed a self-reported symptoms survey. These individuals were included in both analyses. The medical charts and self-reported symptoms of individual cases of COVID-19 patients were reviewed by two physicians.

**Statistical analysis**. To analyze the dynamics of symptoms in COVID-19 cases over time, we plotted the percentages of reported symptoms across different days relative to the time of COVID-19 test. This was considered to be the first positive PCR test result for COVID-19 cases and the first negative PCR test result of negative COVID-19 cases. In addition, for positive COVID-19 cases, we analyzed the prevalence of symptoms relative to the recovery date, which was considered as the date in which a second consecutive negative PCR test result for COVID-19 was recorded. These analyses were done separately for symptoms recorded in primary care visits and those which were self-reported by participants through a survey.

Disease duration was calculated by the number of days from the date of the first positive PCR test to recovery date. Recovery time was modeled with Cox proportional hazards models. Odds ratios (OR) were calculated by a logistic regression model. FDR was employed at the rate of 0.1 on the results. Adjusted OR was further calculated by adjusting for gender, age, chronic medical condition, and time (number of days since study initiation). IPW was applied by fitting a logistic regression model for the probability of being tested (regardless of result). Covariates used for this model were gender, age, chronic medical condition, time (number of days since study initiation), and reported symptoms. The individual probabilities were then used to inversely weight individuals in the logistic regression model.

Time-to-event analyses are presented by Kaplan–Meier curves from the time in which the symptom is self-reported or recorded in EHR to a positive PCR result. Hazard ratios (HR) were calculated by Cox proportional hazards models, adjusted for age, gender, presence of a chronic medical condition and time (number of days since study initiation) to account for the time from symptoms onset to COVID-19 testing results. For self-reported symptoms, IPW was applied to help address biases related to those receiving testing. For both sources of information, two analysis schemes were conducted. In the first scheme, all surveys/ primary care visits are analyzed together, regardless of patient id (i.e., a patient with 4 different surveys will contribute 4 rows to the analysis). In the second scheme, we weighted surveys/ primary care visits by their number per individual (i.e., a patient with 4 different surveys will contribute 4 rows, each with weight ¼ to the analysis). Results are presented in the figures and tables below for the following different types of analysis.

All statistical analyses were performed using Python version 3. Logistic regression was done using the statsmodels package. Time-to-event models were constructed using Python and the lifelines package.

**Ethics declarations**. The study protocol was approved by Maccabi Health Services' institutional review board (0024-20-MHS). Informed consent was waived by the IRB, as all identifying details of the participants were removed before the computational analysis.

**Reporting summary**. Further information on research design is available in the Nature Research Reporting Summary linked to this article.

## Data availability

The data that support the findings of this study originate from Maccabi Health Services. Restrictions apply to the availability of these data and they are therefore not publicly

available. Due to restrictions, these data can be accessed only by request to the authors and/or Maccabi Health Services. Data used for figures is available at https://github.com/barakm-ki/Symptoms-dynamics-of-COVID-19-infection.

## Code availability

Analysis code is available at https://github.com/barakm-ki/Symptoms-dynamics-of-COVID-19-infection though it is tailored to the data and the fields of the Maccabi Health Services database.

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

## Acknowledgements

We thank the following for their contributions to our efforts: Tomer Meir, Amir Gavrieli, Tal Karady, Anastasia Godneva, Saar Shoer, Amit Lavon, Dimitry Kolobkov, Iris Kalka, Ori Cohen, Pini Akiva, Chen Yanover, Guy Amit, Irena Girshovitz, Esma Herzel, Brosh Yinon, Smadar Rain, Sharon Hermoni Alon. H.R. is supported by the Israeli Council for Higher Education (CHE) via the Weizmann Data Science Research Center and by a research grant from Madame Olga Klein–Astrachan.

## Author contributions

These authors contributed equally: Barak Mizrahi, Smadar Shilo, Hagai Rossman. B.M., S.S., and H.R. conceived the project, designed and conducted the analyses, interpreted the results and wrote the manuscript. N.K. conceived and directed the project, designed and conducted the analysis, K.M. designed and conducted the analyses, interpreted the results and wrote the manuscript. N.S.S. and A.E.Z. provided and interpreted data. A.K. interpreted the data. Y.B., V.S., G.C. directed the project, as well as provided the data, E.S. conceived the project, designed and conducted the analyses, interpreted the results and supervised the project and analyses.

## Competing interests

The authors declare no competing interests.
