## [Peer Review File · Nature Communications]

REVIEWER COMMENTS

Reviewer #1 (Remarks to the Author):

In this manuscript, Mizrahi and collaborators investigate the longitudinal prevalence of clinical symptoms in COVID-19 infection diagnosed by PCR testing for SARS-CoV-2 from nasopharyngeal swabs. The authors included data from Electronic Health Record linked to longitudinal self-reported surveys. This allowed them to capture data on symptoms of mostly mild COVID-19 cases from both the patient and the physician perspective.

The other big strengths of the paper are the inclusion of children and the analysis of symptoms dynamics and temporal trend. However, the manuscript is far too long and difficult to follow and the main novel message of the paper (which I believe is summarised in Figures 2 and 3) gets diluted and somehow lost into many secondary confirmatory analyses (eg Figure 4)

I would suggest shortening the manuscript, merging table 1 and 2 into one, moving Fig 1 and 4 to the supplements and concentrating on the longitudinal element of the study.

Also, did the authors adjust for multiple testing when performing multiple comparisons?

Note that ref 9 has been published.

Reviewer #2 (Remarks to the Author):

Summary

Thank you for the opportunity to review this manuscript. The manuscript by Mizrahi, Shilo, and Rossman et al. is a well written manuscript describing the occurrence of symptoms associated with COVID-19 testing among a large dataset derived from the Israeli population combining EHR and self-reported symptom data. A true strength of the data is the prospective and longitudinal nature by which the survey data was collected, however, the authors fail to seize this opportunity and only perform cross-sectional statistical assessments to compliment otherwise largely descriptive data presentation. Without leveraging the strength of this data, the findings are largely incrementally additive/confirmatory with other cross-sectional reports of symptomatic presentation of COVID-19 cited by the authors. The manuscript was poised to make a substantial impact, however, the lack of rigorous statistical methods and even slightly more advanced modeling approaches (e.g. time-to-event analyses) greatly limits the impact of these resultant largely descriptive findings in context of the existing literature cited by the authors. Since the data will not be made publicly available it will be difficult to reproduce these specific findings. Specific comments follow:

Major Comments

1. The manuscript sets the stage that many of the reports to date use hospital data (though that is similarly used here with the complement of self-collected data), retrospectively-collected vs. prospectively-collected, and cross-sectional vs. longitudinal datasets. Of course, there have been several other reports that have used prospective data. While some of this data has been cited, more could be done to more fully characterize the contributions in this area. While this is certainly a weakness of the majority of reports, the authors should highlight/discuss the contributions that have provided data and findings similar to those that they discuss here more comprehensively.
2. Overall the manuscript lacks statistical rigor. The statistical methods do not describe the "Odds-Ratio" models. Presuming these are logistic regression, but there is no discussion of the approach, the models employed, or the statistical software used.
3. As highlighted by the authors, the strength of this dataset is it's prospective and longitudinal collection of data. Thus, it is unclear why the authors have not attempted to perform some type of time-dependent outcome. This is important given the disease course of COVID-19 where symptoms may be differentially present and may bias reports of symptoms. i.e. minor symptoms early on the disease course may be less likely to be reported or result in presentation to the hospital, whereas more serious symptoms, e.g. fever may result in enough concern to report the full spectrum of symptoms / seek treatment. I would prefer to see hazards regression model that

accounts for time to COVID-19 testing results related to symptoms to more sensitively discern the “longitudinal symptom dynamics of COVID-19” as the study is described in the title. Similarly, in the statistical methods section, the authors plot the percentages of reported symptoms across different days relative to the time of the COVID-19 test, since the data is already structured this way, these models would provide a novel aspect upon revision that would greatly improve the strength and impact of the paper.

4. It is not clear in the Odds-Ratio models what, if any, covariates have been adjusted for. Nonetheless, little attempt has been made to adjust for potential confounders for the observed observations significantly limiting the interpretation of the findings. The authors describe in the methods that comorbid condition data was collected this should be considered as potential adjustment variables as these may influence the ability to receive testing or propensity to seek care.
5. Inverse probability weighting or other adjustment methods should be considered to help address biases related to those receiving testing.
6. Given the shifts in testing trends as denoted by the authors, the authors should additionally adjust or stratify for entrance into the cohort according to different time periods.
7. Time to resolution of symptoms could be modeled using a Cox regression model to assess differences rather than the non-parametric Mann-whitney test. A slew of factors (i.e. comorbidities, age, etc.) can contribute to recovery time that seem to be in the dataset, but not used.
8. How were individuals who had not yet recovered considered in the recovery time analysis? How was recovery defined? Were all individuals given the same opportunity to contribute time to “Recovery” in terms of follow-up time?
9. It is not clear why the most robust statistical tests – the odds-ratio analysis was sent to the supplement in favor of largely descriptive figures and tables with no statistical analyses to demonstrate true differences.
10. How was the survey distributed? Was this done using an electronic reporting system? More detail related to survey implementation and the limitations according to survey access should be discussed.
11. A real strength and missed opportunity here is to show validity of self-reported symptoms vs. those reported in the EHR, yet no attempt to demonstrate strong data capture for the 5,083 participants who were similarly recorded in the EHR and via the survey. The authors should take opportunity in a revision to address this and either validate self-reports or clearly demonstrate how self-reports and EHR-captured data may differ.

Minor Comments

1. Figure 1 isn’t completely clear. “Responses that did not meet quality control” seems to overlap with the methods that states individuals who didn’t complete the survey were excluded. Is this the only QC metric?
2. There are a few typographical errors throughout an otherwise well-written manuscript. Most notably there are additional spaces preceding commas in lists in the introduction. Please check a revised manuscript closely for potential errors.

We have addressed the reviewers comments below, with references to parts of the paper that have been modified pursuant to reviewers' suggestions. Our responses are denoted by "**Response**".

REVIEWER COMMENTS

Reviewer #1 (Remarks to the Author):

In this manuscript, Mizrahi and collaborators investigate the longitudinal prevalence of clinical symptoms in COVID-19 infection diagnosed by PCR testing for SARS-CoV-2 from nasopharyngeal swabs. The authors included data from Electronic Health Record linked to longitudinal self-reported surveys. This allowed them to capture data on symptoms of mostly mild COVID-19 cases from both the patient and the physician perspective.

The other big strengths of the paper are the inclusion of children and the analysis of symptoms dynamics and temporal trend.

Comment 1:

However, the manuscript is far too long and difficult to follow and the main novel message of the paper (which I believe is summarized in Figures 2 and 3) gets diluted and somehow lost into many secondary confirmatory analyses (eg Figure 4) I would suggest shortening the manuscript, merging table 1 and 2 into one, moving Fig 1 and 4 to the supplements and concentrating on the longitudinal element of the study.

Response 1:

Thank you for this comment. We have made attempts to shorten the manuscript as suggested and revised the text in order to make the main message more clear. We have moved Figure 1 to the Supplementary appendix. We have merged the two tables and erased less relevant information in them (symptoms which appear in less than 10 positive individuals, reasons for isolation), but kept the information on the two populations separate as the population who responded to the survey and the population who attended primary care visits are different, and have different types of data available. For example, children were not included in the population who filled the surveys and several symptoms were only available for individuals who attended primary care. We believe it is therefore important to show the characteristics of the populations separately for the interpretation of the results by the readers, which are also presented separately for each of the populations.

We believe that presenting the results of the odds ratio analysis in Figure 4 is important and therefore choose to include it in the main text. First, this analysis highlights the most discriminative symptoms for COVID-19 according to our data. Second, as other studies worldwide have performed similar analyses, this allows the comparison of our findings to other countries, for example, Menni et al. 2020 in the UK and USA. Third, to our knowledge, our analysis shows, for the first time, the different OR obtained for different symptoms when recorded by physicians versus self-reported symptoms. In

order to not make this section longer than necessary, we included all the expended results of this analysis in the 3 tables (revised table S3, S4,S5) in section 5 of the supplementary appendix

Comment 2:

Also, did the authors adjust for multiple testing when performing multiple comparisons?

Response 2:

Thank you for this comment. For the comparison between recovery time of different subpopulations, only 3 tests were performed: children versus adults, individuals with chronic medical conditions versus without and males versus females.

For odds ratio analysis of symptoms, we initially did not perform multiple testing corrections. Following your suggestion, we have performed False Discovery Rate (FDR) analysis. This analysis revealed that all self-reported symptoms and primary care recorded symptoms in both children and adults remained significantly associated with positive COVID-19 cases, when FDR was employed at the rate of 0.1. The results are presented in the tables below:

Self-reported symptoms in adults				
	OR	CI	p_value	FDR
Loss of taste and smell	11.18	[6.43-19.44]	1.14E-17	2.16E-16
Confusion	4.02	[1.58-10.21]	0.003	0.013
Headache	2.03	[1.29-3.19]	0.002	0.013
Fatigue	1.73	[1.08-2.79]	0.024	0.075
Fever (Body temperature above 38°C)	1.58	[0.63-3.94]	0.328	0.490
Dry cough	1.52	[0.91-2.54]	0.111	0.302
Diarrhea	1.44	[0.69-3.00]	0.335	0.490
Rhinorrhea and/or nasal	1.42	[0.88-2.29]	0.146	0.309
Cough	1.42	[0.89-2.25]	0.137	0.309

Nausea and/or vomiting	1.32	[0.53-3.28]	0.555	0.659
Wet cough	1.25	[0.69-2.26]	0.470	0.596
Chills	1.01	[0.41-2.52]	0.976	0.976
Myalgia	0.97	[0.50-1.88]	0.920	0.971
Shortness of breath	0.89	[0.36-2.20]	0.794	0.887
Other symptoms	0.71	[0.29-1.77]	0.462	0.596
Sore throat	0.69	[0.37-1.27]	0.232	0.440
Body temperature below 37.4 °C	0.53	[0.34-0.80]	0.003	0.013
No symptoms	0.34	[0.22-0.53]	1.75E-06	1.67E-05
Body temperature between 37.5°C and 37.9°C	0.33	[0.05-2.38]	0.271	0.468

Primary care documented symptoms in adults				
	OR	CI	p_value	FDR
Disturbances of sensation of smell and taste	5.47	[3.69-8.09]	1.89E-17	5.48E-16
Syncope	2.09	[1.13-3.88]	0.018935	0.054911
Runny nose and or nasal congestion	2.09	[1.47-2.95]	3.30E-05	0.000319
Fever	1.62	[1.44-1.83]	6.57E-16	9.52E-15
Speech disturbance	1.37	[0.33-5.69]	0.666374	0.772994

Sleep disturbance	0.94	[0.48-1.84]	0.867606	0.898592
Fatigue	0.89	[0.69-1.14]	0.351065	0.462767
Cough	0.89	[0.78-1.01]	0.074319	0.150954
Headache	0.79	[0.56-1.12]	0.193024	0.282462
Nausea and or vomiting	0.76	[0.42-1.39]	0.371559	0.468487
Myalgia	0.69	[0.55-0.87]	0.001521	0.004939
Emotional disturbance	0.67	[0.52-0.86]	0.001533	0.004939
Dizziness	0.57	[0.30-1.07]	0.07808	0.150954
Abdominal pain	0.53	[0.38-0.75]	0.000277	0.002005
Dyspnea and or Shortness of breath	0.51	[0.26-0.98]	0.043115	0.099555
Diarrhea	0.46	[0.29-0.74]	0.00132	0.004939
Sore throat	0.35	[0.19-0.66]	0.001138	0.004939
Conjunctivitis	0.13	[0.02-0.95]	0.044628	0.099555

	Primary care documented symptoms in children			
	OR	CI	p_value	FDR
Disturbances Of Sensation Of Smell And Taste	2.45	[0.32-18.87]	0.389669	0.596957
Syncope	2.45	[0.58-10.40]	0.223637	0.540402

Emotional Disturbance	2.03	[1.03-4.02]	0.040897	0.163588
Sleep disturbance	1.47	[0.20-10.97]	0.707516	0.786129
Fatigue	1.39	[0.73-2.64]	0.310785	0.565063
Dizziness	1.18	[0.16-8.69]	0.874393	0.874393
Conjunctivitis	1.15	[0.42-3.14]	0.780991	0.822096
Rash	0.81	[0.30-2.21]	0.686773	0.786129
Sore throat	0.66	[0.24-1.79]	0.415215	0.596957
Myalgia	0.64	[0.20-2.02]	0.447717	0.596957
Arthralgia	0.61	[0.08-4.44]	0.626366	0.782957
Headache	0.55	[0.21-1.49]	0.243181	0.540402
Runny nose and or nasal congestion	0.47	[0.12-1.90]	0.28771	0.565063
Speech disturbance	0.45	[0.06-3.26]	0.429671	0.596957
Dyspnea and or Shortness of breath	0.43	[0.16-1.17]	0.097593	0.325309
Cough	0.40	[0.28-0.59]	2.33E-06	2.33E-05
Fever	0.30	[0.22-0.42]	5.27E-13	1.05E-11
Nausea and or vomiting	0.22	[0.03-1.60]	0.135131	0.386089
Diarrhea	0.17	[0.04-0.70]	0.013875	0.092501
Abdominal pain	0.10	[0.01-0.69]	0.019786	0.098928

Comment 3:

Note that ref 9 has been published.

Response 3:

Thank you, we updated the reference accordingly

Reviewer #2 (Remarks to the Author):

Summary

Thank you for the opportunity to review this manuscript. The manuscript by Mizrahi, Shilo, and Rossman et al. is a well written manuscript describing the occurrence of symptoms associated with COVID-19 testing among a large dataset derived from the Israeli population combining EHR and self-reported symptom data. A true strength of the data is the prospective and longitudinal nature by which the survey data was collected, however, the authors fail to seize this opportunity and only perform cross-sectional statistical assessments to compliment otherwise largely descriptive data presentation. Without leveraging the strength of this data, the findings are largely incrementally additive/confirmatory with other cross-sectional reports of symptomatic presentation of COVID-19 cited by the authors. The manuscript was poised to make a substantial impact, however, the lack of rigorous statistical methods and even slightly more advanced modeling approaches (e.g. time-to-event analyses) greatly limits the impact of these resultant largely descriptive findings in context of the existing literature cited by the authors. Since the data will not be made publicly available it will be difficult to reproduce these specific findings. Specific comments follow:

Major Comments

1. The manuscript sets the stage that many of the reports to date use hospital data (though that is similarly used here with the complement of self-collected data), retrospectively-collected vs. prospectively-collected, and cross-sectional vs. longitudinal datasets. Of course, there have been several other reports that have used prospective data. While some of this data has been cited, more could be done to more fully characterize the contributions in this area. While this is certainly a weakness of the majority of reports, the authors should highlight/discuss the contributions that have provided data and findings similar to those that they discuss here more comprehensively.

Response 1:

Thank you for this comment. First, we would like to clarify that the Electronic health records (EHR) analysed in our study originates from the second largest health maintenance organization (HMO) in Israel and contains information from primary care clinics and not hospital's records. We believe that this is a major strength of our study, as most of the reports thus far originated from hospitalized patients. To better clarify

this point, we have highlighted this point in the discussion accordingly. In addition, we have added a more comprehensive discussion regarding the contributions from previous studies on the symptoms in COVID-19 infection.

2. Overall the manuscript lacks statistical rigor. The statistical methods do not describe the “Odds-Ratio” models. Presuming these are logistic regression, but there is no discussion of the approach, the models employed, or the statistical software used.

Response 2:

We apologize for not describing the method used for calculating the odds-ratio. It was indeed calculated by a logistic regression model. All statistical analyses were done using Python 3, and logistic regression was done using the *statsmodels* package. We added a description of this model to the Methods section.

3. As highlighted by the authors, the strength of this dataset is its prospective and longitudinal collection of data. Thus, it is unclear why the authors have not attempted to perform some type of time-dependent outcome. This is important given the disease course of COVID-19 where symptoms may be differentially present and may bias reports of symptoms. i.e. minor symptoms early on the disease course may be less likely to be reported or result in presentation to the hospital, whereas more serious symptoms, e.g. fever may result in enough concern to report the full spectrum of symptoms / seek treatment. I would prefer to see hazards regression model that accounts for time to COVID-19 testing results related to symptoms to more sensitively discern the “longitudinal symptom dynamics of COVID-19” as the study is described in the title. Similarly, in the statistical methods section, the authors plot the percentages of reported symptoms across different days relative to the time of the COVID-19 test, since the data is already structured this way, these models would provide a novel aspect upon revision that would greatly improve the strength and impact of the paper.

Response 3:

We thank you for this important comment. First, we would like to clarify that information on symptoms were obtained in our study by two sources: EHR of primary care visits and self-reported symptoms surveys. Therefore, we are less exposed to the potential bias mentioned which may result from the fact that minor symptoms early on the disease course will be less likely to be reported at hospital admission, as we are not analyzing symptoms at this specific time-point. Nonetheless, we agree with the reviewer on the major importance in performing time to event analyses and as suggested, we have now added time-to-event models to the analysis. We present these analyses by constructing Kaplan-Meier curves from the time in which the symptom is self reported or recorded in the EHR to a positive PCR result. Hazard ratios for each symptom, calculated by Cox proportional hazards models, and adjusted for age, gender, presence of a chronic medical condition and time (number of days since study initiation) are presented to account for the time from symptoms onset to COVID-19 testing results.

We present these results separately for self-reported symptoms and for EHR recorded symptoms. Following the fifth comment given by the reviewer, for self-reported symptoms, Inverse probability weighting (IPW) analysis was performed to help address biases related to those receiving testing (see response 5).

For both sources of information, two analysis schemes were conducted. In the first scheme, all surveys/ primary care visits are analysed together, regardless of patient id (i.e. a patient with 4 different surveys will contribute 4 rows to the analysis). In the second scheme, we weighted surveys/ primary care visits by their number per individual (i.e. a patient with 4 different surveys will contribute 4 rows, each with weight $\frac{1}{4}$ to the analysis). Time-to-event models were constructed using Python and the lifelines package. Results are presented in the figures and tables below for the following different types of analysis.

Overall, these results highlight the importance of loss of taste or smell symptoms throughout the disease course, with an HR of 22.5 and 13.3 followed by fever with HR of 9.7 and 4.3 in self-reported symptoms and EHR respectively. In several symptoms the percentage of positive tests for individuals who reported the symptom or had a record of the symptom in the EHR gradually increased with time, while in others, such as self-reported fever and shortness of breath, a steep increase in the first few days following the symptom occurred. It is most probably due to the fact that the later were part of the testing policy for COVID-19, but also partly since there were a relatively small number of individuals who reported these symptoms. In addition, several symptoms such as nausea and vomiting, muscle pain, headache, and shortness of breath reveal different patterns between the two sources of information. Individuals who self-reported these symptoms had an increasingly higher percentage of positive tests in contrast to their record in the EHR. We added these results to the results section.

1. **Self-reported symptoms** among individuals who were tested for COVID-19 in time. Outcome was considered the first positive PCR test for COVID-19. Negative test results or time after 21 days was considered as censored.
 - a. All surveys

Kaplan-Meier analysis curves for self-reported symptoms of individuals who were tested for COVID-19 in time. The curves present the percentage of individuals who tested positive for COVID-19 and report a specific symptom (red) versus those who did not report this symptom (blue) in time. Hazard ratios (HR) adjusted for gender, age, prior conditions and time (number of days since study initiation) are marked in the figure. All surveys were included in this analysis.

b. Weighted surveys

Kaplan-Meier analysis curves for self-reported symptoms of individuals who were tested for COVID-19 in time. The curves present the percentage of individuals who tested positive for COVID-19 and report a specific symptom (red) versus those who did not report this symptom (blue) in time. Hazard ratios (HR) adjusted for gender, age, prior conditions and time (number of days since study initiation) are marked in figure text. Surveys were weighted according to the number of surveys which were filled by an individual person.

The results of the two analyses schemes for self-reported symptoms are summarized in the table below:

Self - reported symptoms	All surveys		Surveys weighted	
	HR	95% CI	HR	95% CI
Loss of taste and smell	22.49	[18.95-26.70]	13.44	[10.52-17.17]
Fever (body temperature above 38°C)	9.75	[6.07-15.66]	4.57	[2.36-8.85]
Confusion	5.43	[4.00-7.36]	5.12	[3.38-7.77]
Headache	2.6	[2.26-3.00]	2.85	[2.36-3.44]
Dry cough	2.6	[2.23-3.03]	2.02	[1.63-2.52]
Diarrhea	2.54	[2.00-3.22]	3.74	[2.81-4.99]
Fatigue	2.44	[2.04-2.92]	3.03	[2.43-3.78]
Other symptoms	2.3	[1.89-2.80]	1.4	[0.96-2.03]
Chills	2.26	[1.50-3.40]	3.11	[2.02-4.77]
Myalgia	2.01	[1.62-2.50]	1.42	[0.99-2.05]
Rhinorrhea and/or nasal congestion	1.88	[1.66-2.13]	1.59	[1.31-1.93]
cough	1.63	[1.42-1.87]	1.53	[1.26-1.85]
Nausea and/or vomiting	1.61	[1.08-2.40]	2.76	[1.84-4.13]
Sore throat	1.24	[1.00-1.55]	0.86	[0.61-1.22]
Body temperature below 37.4 °C	1.02	[0.93-1.11]	0.82	[0.71-0.95]

Shortness of breath	0.89	[0.60-1.34]	0.56	[0.29-1.06]
Wet cough	0.86	[0.68-1.08]	1.04	[0.79-1.38]
Body temperature between 37.5°C and 37.9°C	0.64	[0.17-2.43]	0.77	[0.2-2.93]
No symptoms	0.48	[0.44-0.52]	0.42	[0.37-0.49]

Table S6: Hazard ratios for each of the self-reported symptoms, calculated by Cox proportional hazards models, and adjusted for age, gender, presence of a chronic medical condition and time (number of days since study initiation) are presented to account for the time from symptoms onset to COVID-19 testing results.

2. **Symptoms documented on EHR** among individuals who tested for COVID-19 in time. Outcome considered was the first positive PCR test for COVID-19

a. All records

Kaplan-Meier analysis curves for EHR recorded symptoms of individuals who were tested for COVID-19. The curves present the percentage of individuals who tested positive for COVID-19 and had documentation of a specific symptom (red) versus those who did not have this symptom (blue) in time. Hazard ratios (HR) adjusted for gender, age, prior conditions and time (number of days since study initiation) are marked in the figure. All records were included in this analysis.

b. Weighted records

Kaplan-Meier analysis curves for EHR recorded symptoms of individuals who were tested for COVID-19. The curves present the percentage of individuals who tested positive for COVID-19 and had documentation of a specific symptom (red) versus those who did not have this symptom (blue) in time. Hazard ratios (HR) adjusted for gender, age, prior conditions and time (number of days since study initiation) are marked in the figure. All records were included in this analysis. Records were weighted according to the number of records per an individual person.

The results of the two analyses schemes for EHR-captured symptoms are summarized in the table below:

Symptom recorded on EHR	All records		Weighted	
	HR	95% CI	HR	95% CI
Disturbances Of Sensation Of Smell And Taste	13.29	[9.56-18.47]	12.17	[8.43-17.57]
Fever	4.32	[3.80-4.90]	4.34	[3.74-5.04]
Fatigue	1.76	[1.41-2.19]	1.81	[1.39-2.35]
Syncope	1.72	[0.99-2.96]	1.95	[1.04-3.66]
Cough	1.42	[1.25-1.61]	1.45	[1.25-1.68]
Runny nose and or nasal congestion	1.38	[1.01-1.90]	1.29	[0.90-1.85]
Diarrhea	1.01	[0.66-1.53]	0.89	[0.54-1.48]
Sore throat	0.87	[0.51-1.48]	0.78	[0.43-1.43]
Headache	0.85	[0.62-1.16]	0.89	[0.62-1.28]
Nausea and or vomiting	0.81	[0.48-1.37]	0.68	[0.34-1.38]
Sleep disturbance	0.81	[0.46-1.43]	0.73	[0.34-1.56]
Chest Pain or discomfort	0.65	[0.47-0.91]	0.5	[0.32-0.77]
Dizziness	0.64	[0.38-1.07]	0.51	[0.25-1.04]
Emotional Disturbance	0.59	[0.48-0.73]	0.64	[0.49-0.84]
Abdominal pain	0.5	[0.37-0.67]	0.43	[0.29-0.62]

Arthralgia	0.46	[0.33-0.64]	0.39	[0.25-0.59]
Myalgia	0.46	[0.37-0.57]	0.44	[0.34-0.57]
Dyspnea and or Shortness of breath	0.45	[0.26-0.78]	0.57	[0.31-1.04]

Table S7: Hazard ratios for each of the EHR-captured symptom, calculated by Cox proportional hazards models, and adjusted for age, gender, presence of a chronic medical condition and time (number of days since study initiation) are presented to account for the time from symptoms onset to COVID-19 testing results.

We added a section in the results which describes these results. Kaplan-Meier with the result obtained for all surveys and all EHR and adjusted HR values were added to the revised Figure 1. Full results were added to section 6 of the Supplementary appendix (tables S6 and S7).

Figure 1. Dynamics of symptoms in COVID-19 patients. columns **A-C**: each row represents the prevalence of a symptom in our cohort analysed by time relative to diagnosis day from **A**: Survey of self-reported symptoms and **B**: Primary care visits in positive COVID19 cases (red) versus negative (blue) **C** Prevalence of symptoms in our cohort relative to time of recovery by surveys of self-reported symptoms (purple) and primary care visits (green). Each time point is calculated by taking a 1-week window (± 3 days from day). Columns **D-E**: Kaplan-Meier curves from the time in which a symptom is self-reported (**D**) or recorded in the EHR (**E**) to a positive PCR result. The curves present the cumulative incidence of individuals who tested positive

for COVID-19 and have a specific symptom (orange) versus those who did not report this symptom (purple) in time. Hazard ratios (HR) adjusted for gender, age, prior conditions and time (number of days since study initiation) are indicated. Note y-axis scale is different for each panel.

4. It is not clear in the Odds-Ratio models what, if any, covariates have been adjusted for. Nonetheless, little attempt has been made to adjust for potential confounders for the observed observations significantly limiting the interpretation of the findings. The authors describe in the methods that comorbid condition data was collected this should be considered as potential adjustment variables as these may influence the ability to receive testing or propensity to seek care.

Response 4:

Thank you for suggesting that. We have adjusted the odds ratio model by the following covariates: age, gender, presence of a chronic medical condition and time (number of days since study initiation). Symptoms that were previously reported in the manuscript as having a significant OR for COVID-19 disease using the basic model- Loss of taste or smell, Confusion, Headache and Fatigue, also had a significant OR in the adjusted model. We have added a table to section 5 of the supplementary appendix with the full results of this analysis. These results are specified below:

A. Self reported symptoms in adults

Self-reported symptoms in adults	Basic Model		Adjusted Model	
	OR	CI	OR	CI
Loss of taste and smell	11.18	[6.43-19.44]	9.34	[5.26-16.58]
Confusion	4.02	[1.58-10.21]	3.14	[1.21-8.18]
Headache	2.03	[1.29-3.19]	1.83	[1.15-2.90]
Fatigue	1.73	[1.08-2.79]	1.72	[1.06-2.80]
Fever (Body temperature above 38°C)	1.58	[0.63-3.94]	1.48	[0.59-3.74]
Dry cough	1.52	[0.91-2.54]	1.18	[0.70-2.00]
Diarrhea	1.44	[0.69-3.00]	1.34	[0.64-2.83]
Rhinorrhea and/or nasal	1.42	[0.88-2.29]	1.23	[0.76-1.99]

Cough	1.42	[0.89-2.25]	1.08	[0.67-1.72]
Nausea and/or vomiting	1.32	[0.53-3.28]	1.17	[0.46-2.95]
Wet cough	1.25	[0.69-2.26]	0.99	[0.54-1.81]
Chills	1.01	[0.41-2.52]	0.92	[0.37-2.30]
Myalgia	0.97	[0.50-1.88]	1	[0.51-1.95]
Shortness of breath	0.89	[0.36-2.20]	0.73	[0.29-1.83]
Other symptoms	0.71	[0.29-1.77]	0.9	[0.36-2.26]
Sore throat	0.69	[0.37-1.27]	0.65	[0.35-1.22]
Body temperature below 37.4 °C	0.53	[0.34-0.80]	0.68	[0.44-1.05]
No symptoms	0.34	[0.22-0.53]	0.51	[0.32-0.80]
Body temperature between 37.5°C and 37.9°C	0.33	[0.05-2.38]	0.36	[0.05-2.62]

Odds ratio calculation for self-reported symptoms in adults 21 days prior to the date of diagnosis. Adjusted model takes into account the following covariates: age, gender, presence of a chronic medical condition and time (number of days since study initiation). For positive COVID19 cases, this date was considered as the first positive PCR test. For COVID negative cases, this test was considered as the first negative result for COVID-19.

These results are also presented in the revised Figure 3, along with the results of IPW analysis (see response5).

Figure 3: Odds ratio analysis of symptoms prior to COVID-19 test. Log odds ratio (OR) calculated by the prevalence of symptoms 21 days prior to COVID-19 test for adults in self recorded symptoms versus EHR-captured symptoms. Grey lines indicate 95% confidence intervals. Blue circles represent log OR calculated from the basic model, orange circles represent log OR from the adjusted model, green circles represent log OR after applying Inverse probability weighting (IPW) analysis to help address biases related to those receiving testing and red circles represent log OR from the adjusted model after applying IPW.

B. Symptoms documented in EHR of adults

Primary care documented symptoms in adults	Basic model		Adjusted model	
	OR	CI	OR	CI
Disturbances of sensation of smell and taste	5.47	[3.69-8.09]	6.83	[4.51-10.33]
Syncope	2.09	[1.13-3.88]	2.52	[1.33-4.77]
Runny nose and or nasal congestion	2.09	[1.47-2.95]	1.9	[1.33-2.72]

Fever	1.62	[1.44-1.83]	1.66	[1.47-1.87]
Speech disturbance	1.37	[0.33-5.69]	1.17	[0.27-5.03]
Sleep disturbance	0.94	[0.48-1.84]	1.12	[0.57-2.21]
Fatigue	0.89	[0.69-1.14]	1.17	[0.91-1.52]
Cough	0.89	[0.78-1.01]	0.79	[0.69-0.90]
Headache	0.79	[0.56-1.12]	0.87	[0.61-1.24]
Nausea and or vomiting	0.76	[0.42-1.39]	0.82	[0.45-1.50]
Myalgia	0.69	[0.55-0.87]	0.76	[0.60-0.97]
Emotional disturbance	0.67	[0.52-0.86]	0.79	[0.61-1.01]
Dizziness	0.57	[0.30-1.07]	0.74	[0.39-1.39]
Abdominal pain	0.53	[0.38-0.75]	0.59	[0.42-0.83]
Dyspnea and or Shortness of breath	0.51	[0.26-0.98]	0.42	[0.22-0.82]
Diarrhea	0.46	[0.29-0.74]	0.69	[0.43-1.10]
Sore throat	0.35	[0.19-0.66]	0.44	[0.24-0.83]
Conjunctivitis	0.13	[0.02-0.95]	0.11	[0.02-0.77]

C. Symptoms documented in EHR of children

Primary care documented symptoms in children	Basic model		Adjusted Model	
	OR	CI	OR	CI
Disturbances Of Sensation Of Smell And Taste	2.45	[0.32-18.87]	2.26	[0.28-18.54]
Syncope	2.45	[0.58-10.40]	1.94	[0.42-8.86]
Emotional Disturbance	2.03	[1.03-4.02]	1.60	[0.79-3.24]
Sleep disturbance	1.47	[0.20-10.97]	1.84	[0.23-14.77]
Fatigue	1.39	[0.73-2.64]	1.33	[0.69-2.56]
Dizziness	1.18	[0.16-8.69]	1.00	[0.13-7.67]
Conjunctivitis	1.15	[0.42-3.14]	1.17	[0.42-3.26]
Rash	0.81	[0.30-2.21]	1.06	[0.38-2.93]
Sore throat	0.66	[0.24-1.79]	0.66	[0.24-1.81]
Myalgia	0.64	[0.20-2.02]	0.54	[0.17-1.72]
Arthralgia	0.61	[0.08-4.44]	0.58	[0.08-4.32]
Headache	0.55	[0.21-1.49]	0.39	[0.14-1.07]
Runny nose and or nasal congestion	0.47	[0.12-1.90]	0.53	[0.13-2.21]
Speech disturbance	0.45	[0.06-3.26]	0.57	[0.08-4.26]
Dyspnea and or Shortness of breath	0.43	[0.16-1.17]	0.50	[0.18-1.38]
Cough	0.40	[0.28-0.59]	0.36	[0.25-0.53]

Fever	0.30	[0.22-0.42]	0.34	[0.24-0.47]
Nausea and or vomiting	0.22	[0.03-1.60]	0.20	[0.03-1.42]
Diarrhea	0.17	[0.04-0.70]	0.22	[0.05-0.87]
Abdominal pain	0.10	[0.01-0.69]	0.09	[0.01-0.61]

Table S3: Odds ratio calculation for symptoms in children that were documented in primary care visits 21 days prior to the date of diagnosis. Adjusted model takes into account the following covariates: age, gender, presence of a chronic medical condition and time (number of days since study initiation). For positive COVID19 cases, this date was considered as the first positive PCR test. For COVID negative cases, this test was considered as the first negative result for COVID-19.

These results are also presented in supplementary Figure S3:

5. *Inverse probability weighting or other adjustment methods should be considered to help address biases related to those receiving testing.*

Response 5:

Thank you for this excellent suggestion, we have now included IPW analysis to help address biases related to those receiving testing.

Inverse Probability weighting was applied by fitting a logistic regression model for the probability of being tested (regardless of result). Covariates used for this model were gender, age, chronic medical condition, time (number of days since study initiation) and reported symptoms. The individual probabilities were then used to inversely weight individuals in the logistic regression model. The results of this analysis are now described in the manuscript and below, alongside the results of the OR obtained by the basic and adjusted logistic regression models. When applying IPW and the adjusted IPW, all self-reported symptoms which were significant by the basic and adjusted OR model, remain significant with the exception of fatigue. Of note, IPW was applied only for self-reported symptoms, as the population of individuals who had information from primary care visits was available only for individuals tested for COVID-19.

Self reported symptoms in adults	Basic Model		Adjusted Model		IPW		IPW Adjusted	
	OR	CI	OR	CI	OR	CI	OR	CI
Loss of taste and smell	11.18	[6.43-19.44]	9.34	[5.26-16.58]	15.46	[6.02-39.74]	12.67	[4.79-33.46]
Confusion	4.02	[1.58-10.21]	3.14	[1.21-8.18]	4.44	[1.16-16.96]	3.71	[0.95-14.41]
Headache	2.03	[1.29-3.19]	1.83	[1.15-2.90]	2.08	[1.1-3.94]	1.91	[1-3.64]
Fatigue	1.73	[1.08-2.79]	1.72	[1.06-2.80]	1.85	[0.83-4.09]	1.83	[0.82-4.09]
Fever (Body temperature above 38°C)	1.58	[0.63-3.94]	1.48	[0.59-3.74]	2.13	[0.15-30.17]	2.05	[0.14-29.55]
Dry cough	1.52	[0.91-2.54]	1.18	[0.70-2.00]	1.68	[0.73-3.87]	1.34	[0.58-3.11]
Diarrhea	1.44	[0.69-3.00]	1.34	[0.64-2.83]	2.42	[0.96-6.15]	2.26	[0.88-5.76]
Rhinorrhea and/or nasal congestion	1.42	[0.88-2.29]	1.23	[0.76-1.99]	1.47	[0.79-2.75]	1.29	[0.68-2.42]

Cough	1.42	[0.89-2.25]	1.08	[0.67-1.72]	1.37	[0.68-2.77]	1.07	[0.53-2.18]
Nausea and/or vomiting	1.32	[0.53-3.28]	1.17	[0.46-2.95]	1.78	[0.43-7.44]	1.55	[0.37-6.54]
Wet cough	1.25	[0.69-2.26]	0.99	[0.54-1.81]	1.16	[0.44-3.1]	0.92	[0.34-2.46]
Chills	1.01	[0.41-2.52]	0.92	[0.37-2.30]	2.18	[0.46-10.42]	1.97	[0.41-9.52]
Myalgia	0.97	[0.50-1.88]	1	[0.51-1.95]	0.8	[0.22-2.98]	0.82	[0.22-3.05]
Shortness of breath	0.89	[0.36-2.20]	0.73	[0.29-1.83]	0.94	[0.18-4.86]	0.79	[0.15-4.12]
Other symptoms	0.71	[0.29-1.77]	0.9	[0.36-2.26]	0.91	[0.29-2.88]	1	[0.32-3.17]
Sore throat	0.69	[0.37-1.27]	0.65	[0.35-1.22]	0.63	[0.18-2.17]	0.59	[0.17-2.06]
Body temperature below 37.4 °C	0.53	[0.34-0.80]	0.68	[0.44-1.05]	0.55	[0.34-0.89]	0.66	[0.41-1.07]
No symptoms	0.34	[0.22-0.53]	0.51	[0.32-0.80]	0.43	[0.27-0.69]	0.53	[0.33-0.86]
Body temperature between 37.5°C and 37.9°C	0.33	[0.05-2.38]	0.36	[0.05-2.62]	0.22	[0-103.13]	0.22	[0-105.68]

Table S4: Odds ratio calculation for symptoms in Adults that were documented in primary care visits 21 days prior to the date of congestion diagnosis. Adjusted model takes into account the following covariates: age, gender, presence of a chronic medical condition and time (number of days since study initiation). Inverse Probability weighting (IPW) was applied by fitting a logistic regression model for the probability of being tested (regardless of result). For positive COVID19 cases, this date was considered as the first positive PCR test. For COVID negative cases, this test was considered as the first negative result for COVID-19.

Figure 3: Odds ratio analysis of symptoms prior to COVID-19 test. Log odds ratio (OR) calculated by the prevalence of symptoms 21 days prior to COVID-19 test for adults in self recorded symptoms versus EHR-captured symptoms. Grey lines indicate 95% confidence intervals. Blue circles represent log OR calculated from the basic model, orange circles represent log OR from the adjusted model, green circles represent log OR after applying Inverse probability weighting (IPW) analysis to help address biases related to those receiving testing and red circles represent log OR from the adjusted model after applying IPW

6. Given the shifts in testing trends as denoted by the authors, the authors should additionally adjust or stratify for entrance into the cohort according to different time periods.

Response 6:

We fully agree with your comment and adjusted our OR analysis also by the number of days since the initiation of the study.

7. Time to resolution of symptoms could be modeled using a Cox regression model to assess differences rather than the non-parametric Mann-Whitney test. A slew of factors (i.e. comorbidities, age, etc.) can contribute to recovery time that seem to be in the dataset, but not used.

Response 7:

Thank you for this comment. We wish to clarify that we used Mann-Whitney test to compare recovery time between different groups of individuals. Recovery time was considered as the date in which a second consecutive negative PCR test result for COVID-19 was recorded, in line with the Israeli ministry of health policy during the study period. Unfortunately, modeling the time of symptoms resolution was not feasible

based on our data, as the majority of individuals did not fulfill the survey every day, and therefore it was not possible to infer the exact timing of the resolution of a specific symptom.

Following your suggestion, we modeled the recovery time by Cox regression models. We agree that many factors may contribute to recovery time and thus used these models to further analyse the effect of age, gender and the presence of comorbidities on recovery time. These analyses revealed that children have a significantly shorter recovery time compared to adults ($p=0.04$). Gender and the presence of a chronic medical condition did not affect the recovery time significantly ($p=0.46$ and $p=0.87$ respectively). These analyses are presented in the following figures and table:

Covariate	HR	95% CI	p-value
Presence of a chronic medical condition	0.99	[0.91-1.08]	0.87
Female	1.03	[0.95-1.13]	0.46
Children	1.18	[1.01-1.39]	0.04

8. How were individuals who had not yet recovered considered in the recovery time analysis? How was recovery defined? Were all individuals given the same opportunity to contribute time to “Recovery” in terms of follow-up time?

Response 8:

Recovery date was considered as the date in which a second consecutive negative PCR test result for COVID-19 was recorded (see Methods). Analysis of recovery time was calculated on a subsample of the cohort who recovered by these criteria. Only recovered individuals took part in this analysis. The duration of the disease was calculated by the

number of days from the date of the first positive PCR test to recovery date for the 2,045 recovered patients in our data. Their recovery time is detailed in the following table:

population	Number of patients (prev numbers)	Mean (+-SD) of time to recovery
all	2,045	23.5+-9.9
Children	177	21.7+-8.8
Adults	1,868	23.7+-9.9
Male	1,093	23.3+-9.8
Female	952	23.8+-9.9

We agree with the reviewer comment that a better modeling approach is to analyse recovery time using cox regression models, which can model censoring properly. We therefore performed these analyses which are presented above. In this analysis, all positive patients, including patients who did not yet recover as they did not have enough follow up time were included, and censored at the last time of data available

9. It is not clear why the most robust statistical tests – the odds-ratio analysis was sent to the supplement in favor of largely descriptive figures and tables with no statistical analyses to demonstrate true differences.

Response 9:

We fully agree that the odds-ratio analysis is of major importance and therefore included the results of this analysis in the main text and presented them visually in Figure 4. In addition, we added 3 tables (table S2, S3,S4) that describe the full results in section 5 of the supplementary appendix

10. How was the survey distributed? Was this done using an electronic reporting system? More detail related to survey implementation and the limitations according to survey access should be discussed.

Response 10:

Thank you for this comment. One of the advantages of our study is that it relies on the responses from a survey distributed by Maccabi health system (MHS), Israeli second largest Health maintenance organization (HMO). A link to the electronic survey was distributed by emails and text messages to all the adult members of MHS . In Israel, participation in a medical insurance plan is compulsory and all residents are entitled to basic health care as a fundamental right. The Israeli healthcare system is based on the National Health Insurance Law of 1995, which mandates all citizens resident in the country to join one of four official HMO which are run as not-for-profit organizations

and are prohibited by law from denying any Israeli resident membership. Moreover, every Israeli resident has a right to change their HMO. Therefore, the distribution of the survey by the HMO to all the adult members is representative to the Israeli population and includes individuals from different and representative socioeconomic backgrounds. To better clarify this point, we revised the relevant section in the Methods and the Discussion.

11. A real strength and missed opportunity here is to show validity of self-reported symptoms vs. those reported in the EHR, yet no attempt to demonstrate strong data capture for the 5,083 participants who were similarly recorded in the EHR and via the survey. The authors should take opportunity in a revision to address this and either validate self-reports or clearly demonstrate how self-reports and EHR-captured data may differ.

Response 11:

In order to check the validity of our survey, we directly compared self-reported answers on COVID-19 diagnoses in our survey to the documented test in the medical chart and found a very high agreement between the two sources. This point is discussed in the discussion. For example, from those who reported not diagnosed with COVID-19 in the survey, only 0.03% had a record of a positive COVID-19 test in the EHR thus further validating our survey.

We agree with the reviewer that it is also an opportunity to validate self-reports of symptoms or demonstrate how self-reports and EHR-captured data may differ. However, the comparison between self-reported symptoms and EHR recorded symptoms is more problematic, as symptoms may be dynamic and the timing of the visits in the clinics and the survey filling are not identical. To further investigate this issue, we searched for individuals among the 5,083 participants who filled a symptom survey on the same date in which they had a clinic visit. In total, we identified 915 different events in which the same person filled a self-reported survey and had a physician visit documented in the EHR on the same day, for a total of 706 different individuals. When comparing these events, we found the overall agreement between the two sources was generally low. Overall, most of the symptoms, with the exception of fever and myalgia, were self-reported in a higher percentage than they were recorded by physicians in the EHR. The results of the comparison are presented in the table below. As expected, symptoms which are part of the Israeli testing policy had a higher agreement between the two sources since they were more likely to be asked by a physician during the visit. These included cough, which had a 52% agreement between the two sources and fever, which had a 34% agreement. Diarrhea also had a relatively high agreement of 35%. Other symptoms had a lower agreement of up to 16%. Disturbance of the sensation in smell and taste, had no agreement at all between the two sources, and were mostly self-reported, potentially due to the fact that early in the course of the pandemic the evidence on the existence of this symptoms in individuals

infected with COVID-19 was not strong, so it is possible that it was less asked and reported by physicians.

The low agreement between both data sources may be due to several reasons including the local practices of diagnosis documentation by physicians, who usually document only the main diagnoses as ICD-9 codes, as there is limited time dedicated to the clinic visit. Moreover, previous studies comparing individual's self-report versus clinicians reporting in regard to other diagnosis, have found that physicians are more likely to record diagnoses for more severe, complex cases (Morgan, Maria A., et al. General hospital psychiatry, 2019) and that self-report is more sensitive to identifying symptoms-based conditions (VIOLÁN, Concepción, et al. BMC public health, 2013). Altogether, we believe that this point highlights the strength of our study in integrating two different data sources together, each with its own pros and cons, to obtain a more complete picture on symptoms as reflected by both perspectives.

We added a section on this analysis to the results and discussed it in the discussion. The full results were added as Table S1 to section 1 of the supplementary appendix:

Symptom	total	Symptom appears in both data sources	Symptom was only recorded in a primary care visit	Symptom was only self-reported in the survey
Cough	393	206 (52%)	70 (18%)	117 (30%)
Diarrhea	103	36 (35%)	20 (19%)	47 (46%)
Disturbances Of Sensation Of Smell And Taste	32	0 (0%)	5 (16%)	27 (84%)
Dyspnea and or Shortness of breath	116	5 (4%)	11 (9%)	100 (86%)
Fatigue	312	51 (16%)	44 (14%)	217 (70%)
Fever	165	56 (34%)	99 (60%)	10 (6%)
Headache	248	34 (14%)	19 (8%)	195 (79%)

Myalgia	311	34 (11%)	139 (45%)	138 (44%)
Nausea and or vomiting	63	2 (3%)	9 (14%)	52 (83%)
Runny nose and or nasal congestion	193	6 (3%)	8 (4%)	179 (93%)
Sore throat	202	18 (9%)	13 (6%)	171 (85%)

Table S1: comparison between self-reported symptoms and Electronic health record-captured data.

Minor Comments

1. Figure 1 isn't completely clear. "Responses that did not meet quality control" seems to overlap with the methods that states individuals who didn't complete the survey were excluded. Is this the only QC metric?

Response 12:

Indeed, the only QC metric was surveys which were not fully completed by participants and were therefore excluded from the analysis. For clarity, we added it to the legend of this figure which was moved to the supplementary in response to a comment by the first reviewer and is now termed Figure S1.

2. There are a few typographical errors throughout an otherwise well-written manuscript. Most notably there are additional spaces preceding commas in lists in the introduction. Please check a revised manuscript closely for potential errors.

Response 13:

Thank you very much, we apologize for the typographical errors. We have erased the additional spaces, revised the manuscript again and corrected additional errors.

REVIEWERS' COMMENTS

Reviewer #1 (Remarks to the Author):

The authors addressed all my comments

Reviewer #2 (Remarks to the Author):

Thank you for the opportunity to review the resubmission of this manuscript. By addressing all of the comments put forth by myself and the other reviewers, the manuscript has been greatly improved and, in my opinion, increases the potential impact. I only have minor comments that should be addressed prior to publication.

Minor Comments

1. Abstract: Objectives: Suggest removing the line "most studies thus far are based on hospitalized patients" at this stage in the game enough prospective reports of general populations have been published that the authors can instead highlight instead that rarely do cohorts have access to EHR (hospitalized) data, data from primary care settings, AND longitudinal prospective data on symptoms. This will provide space to address comment 3.
2. Abstract: Design section: Remove the word "linked" as this doesn't apply to ALL cases.
3. Part of the novelty of this report is the discordance between EHR and survey data, and as the authors point out that survey data may more sensitively capture symptom data. Reference to this could be added to the abstract under the results or conclusion headers. Thank you for performing this secondary analysis – this illustrated a very important point for the field going forward.
4. The symptom lists in Parts A and B of Table 1 should be similarly ordered (alphabetical) for more clear direct comparison, or alternatively ranked according to prevalence.
5. The conclusion may more explicitly state that there is a limitation associated with using EHR data compared to survey data in capturing the "full spectrum" of symptoms due to clinical practice not querying ALL symptoms vs. presenting these options in a survey. This uniquely highlights the power of this and other survey derived data in addressing this pandemic compared to typical survey studies.
6. There remain several typographical errors throughout the manuscript. E.g. Line 76 ", Information regarding dynamic..." should be ". Information regarding dynamics" and other hanging spaces scattered throughout the manuscript that I expect will be corrected before publication.

We have addressed the reviewers comments below, with references to parts of the paper that have been modified pursuant to reviewers' suggestions. Our responses are denoted by "**Response**".

REVIEWERS' COMMENTS

Reviewer #1 (Remarks to the Author):

The authors addressed all my comments

Reviewer #2 (Remarks to the Author):

Thank you for the opportunity to review the resubmission of this manuscript. By addressing all of the comments put forth by myself and the other reviewers, the manuscript has been greatly improved and, in my opinion, increases the potential impact. I only have minor comments that should be addressed prior to publication.

Minor Comments

Comment 1:

1. Abstract: Objectives: Suggest removing the line "most studies thus far are based on hospitalized patients" at this stage in the game enough prospective reports of general populations have been published that the authors can instead highlight instead that rarely do cohorts have access to EHR (hospitalized) data, data from primary care settings, AND longitudinal prospective data on symptoms. This will provide space to address comment 3.

Response 1:

Thank you for this comment. We note that the EHR analysed in our study were from primary care visits. We have removed this sentence and revised according to your suggestions.

Comment 2:

2. Abstract: Design section: Remove the word "linked" as this doesn't apply to ALL cases.

Response 2:

This word was removed from the abstract.

Comment 3:

3. Part of the novelty of this report is the discordance between EHR and survey data, and as the authors point out that survey data may more sensitively capture symptom data. Reference to this could be added to the abstract under the results or conclusion headers. Thank you for performing this secondary analysis – this illustrated a very important point for the field going forward.

Response 3:

We greatly appreciate your suggestion to perform these analyses and believe they have greatly improved the paper. We added these results to the abstract.

Comment 4:

4. *The symptom lists in Parts A and B of Table 1 should be similarly ordered (alphabetical) for more clear direct comparison, or alternatively ranked according to prevalence.*

Response 4:

We reordered the symptoms in the table according to alphabetical order.

Comment 5:

5. *The conclusion may more explicitly state that there is a limitation associated with using EHR data compared to survey data in capturing the “full spectrum” of symptoms due to clinical practice not querying ALL symptoms vs. presenting these options in a survey. This uniquely highlights the power of this and other survey derived data in addressing this pandemic compared to typical survey studies.*

Response 5:

We thank you for this comment and added this point to the conclusion.

Comment 6:

6. *There remain several typographical errors throughout the manuscript. E.g. Line 76 “, Information regarding dynamic...” should be “. Information regarding dynamics” and other hanging spaces scattered throughout the manuscript that I expect will be corrected before publication.*

Response 6:

Thank you very much, we apologize for the typographical errors. We have erased the additional spaces, revised the manuscript again and corrected additional errors.